# Insulin signaling controls neurotransmission via the 4eBP-dependent modification of the exocytotic machinery

Rebekah Elizabeth Mahoney[1,2], Jorge Azpurua[1], Benjamin A Eaton[1,2]*

[1]Department of Physiology, University of Texas Health Sciences Center at San Antonio, San Antonio, United States; [2]Barshop Institute of Aging and Longevity Studies, University of Texas Health Sciences Center at San Antonio, San Antonio, United States

**Abstract** Altered insulin signaling has been linked to widespread nervous system dysfunction including cognitive dysfunction, neuropathy and susceptibility to neurodegenerative disease. However, knowledge of the cellular mechanisms underlying the effects of insulin on neuronal function is incomplete. Here, we show that cell autonomous insulin signaling within the *Drosophila* CM9 motor neuron regulates the release of neurotransmitter via alteration of the synaptic vesicle fusion machinery. This effect of insulin utilizes the FOXO-dependent regulation of the *thor* gene, which encodes the *Drosophila* homologue of the eif-4e binding protein (4eBP). A critical target of this regulatory mechanism is Complexin, a synaptic protein known to regulate synaptic vesicle exocytosis. We find that the amounts of Complexin protein observed at the synapse is regulated by insulin and genetic manipulations of Complexin levels support the model that increased synaptic Complexin reduces neurotransmission in response to insulin signaling.

*For correspondence: eatonb@uthscsa.edu

**Competing interests:** The authors declare that no competing interests exist.

## Introduction

Metabolic disorders such as diabetes are associated with widespread declines in neuronal function including peripheral and proximal neuropathy, retinopathy, reduced cognition, impaired motor functions and increased risk of developing neurodegenerative disease including Alzheimer's disease (*Deak and Sonntag, 2012*; *Gispen and Biessels, 2000*; *Luchsinger, 2012*; *Park, 2001*; *Plum et al., 2005*). The loss of normal synapse function is believed to be an important contributor to all these disorders suggesting that changes in insulin signaling can influence synaptic connectivity throughout the nervous system. For example, analysis of human patients with type II diabetes (T2DM) reveals changes in brain structures, including synapse numbers, which correlate with decreased cognitive performance (*Qiu et al., 2014*). In addition, numerous rodent studies have demonstrated that changes in peripheral and cerebral insulin result in changes to synapse function and plasticity in both the hippocampus and retinae (*Gispen and Biessels, 2000*; *Hombrebueno et al., 2014*). Rodent and human studies have also demonstrated that changes in normal insulin signaling can alter peripheral synapses including neuromuscular junctions (NMJs) (*Allen et al., 2015a*, *2015b*; *Fahim et al., 1998*; *Francis et al., 2011*; *Garcia et al., 2012*; *Ramji et al., 2007*). Despite the wide-spread effects of altered insulin signaling on synapse function, the cellular mechanisms underlying the effects insulin signaling on synapse function, especially the control of neurotransmitter release, are poorly understood.

**eLife digest** The rates of obesity and diabetes are increasing worldwide. Both conditions produce a wide range of detrimental effects on health, including an increased risk of developing neurodegenerative diseases such as Alzheimer's disease.

Obesity and diabetes reduce how well many of the body's cells can respond to a hormone called insulin. Insulin signaling is believed to influence how the brain works, but this had not been studied in detail. Mahoney et al. have now studied the fruit fly *Drosophila melanogaster* to investigate whether insulin signaling within neurons can directly alter neurotransmission – the process by which neurons communicate with each other by releasing chemicals called neurotransmitters.

The fruit flies were fed a high protein diet, which increased their insulin signaling and reduced the activity of a protein called FOXO in the neurons. This resulted in the reduced transcription of the translational inhibitor 4eBP and ultimately caused an increase in the amount of the Complexin protein. This protein in turn reduced the release of neurotransmitters. Thus, the results of the experiments demonstrate that insulin signaling within adult fruit fly neurons decreases neurotransmission.

Future experiments will be needed to study these mechanisms in more detail. One of the remaining open questions is where proteins such as Complexin are being made in the neuron.

There exist well-established evolutionarily conserved targets of insulin signaling that have been implicated in the effects of insulin on synapse function (*Kleinridders et al., 2014*; *Park, 2001*; *Plum et al., 2005*). This includes the mammalian target of rapamycin (mTOR) complex that is positively regulated by insulin signaling. In the postsynaptic compartment, TOR signaling has been directly implicated in the regulation of post-synaptic function including the formation of new synapses and the generation of retrograde signaling during homeostatic synaptic plasticity (*Penney et al., 2012*; *Stoica et al., 2011*; *Takei and Nawa, 2014*; *Weston et al., 2012*). The role of TOR signaling within the presynaptic nerve terminal is less clear. Another important target of insulin signaling is the FOXO family of transcription factors. Insulin negatively regulates FOXO via phosphorylation by Akt in both flies and rodents (*Puig et al., 2003*; *Teleman et al., 2005*; *Yamamoto and Tatar, 2011*). Previous studies have established that FOXO is required in *Drosophila* larval motor neurons for synapse growth, synaptic vesicle recycling, and for the control of neuronal excitability downstream of PI3K signaling (*Howlett et al., 2008*; *Nechipurenko and Broihier, 2012*). In mammals, recent studies have revealed a requirement for FOXO6, a FOXO family member highly expressed in the hippocampus, during learning and memory (*Salih et al., 2012*). It was shown in these studies that FOXO6 was required for the expression of genes involved in neurotransmission supporting a direct role for FOXO in the regulation of synapse function (*Salih et al., 2012*). It is unclear whether insulin signaling regulates FOXO activity in neurons in any system. In the present study, we present evidence that in adult Drosophila motor neurons, insulin signaling negatively regulates the presynaptic release of neurotransmitter via the FOXO-dependent regulation of the translational inhibitor the eukaryotic initiation factor 4e binding protein (4eBP). The translational target of this signaling system appears to be the Complexin protein, which is known to regulate the exocytosis of synaptic vesicles providing direct link between neuronal insulin signaling and neurotransmitter release.

## Results

### Dietary protein regulates the probability of release at the CM9 NMJ

Electrical recordings from the CM9 muscle group on the adult fly proboscis previously revealed a decrease in the amount of neurotransmitter released from the CM9 NMJ in flies raised on a high-calorie diet compared to flies raised on a low-calorie diet (*Rawson et al., 2012*). The CM9 NMJ is ideal for this research since it combines motor neuron-specific genetic manipulations with a robust synaptic recording preparation allowing us to interrogate the cell autonomous effects of diet (*Figure 1A*). Using this system, we have extended our initial observations by first determining that changing only the yeast component of the diet is sufficient to alter neurotransmission. Animals raised for 21 days

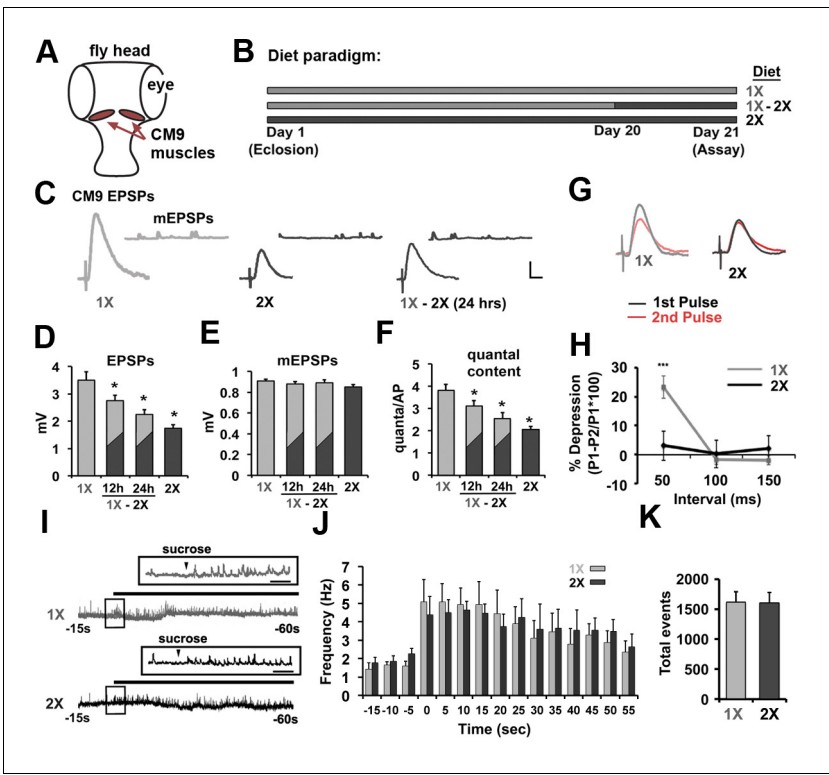

**Figure 1.** Effects of dietary protein concentrations on neurotransmission at the CM9 NMJ. (A) Diagram of Drosophila head indicating the location of the Cibarial Muscle 9 (CM9). (B) 21-day feeding paradigm used for the analysis of dietary effects on neurotransmission. Animals were raised for 21 days on a low-protein diet (1X = light gray), a high-protein diet (2X = dark gray), or subjected to a shift from a low-protein diet (1X) to a high-protein diet (2X) on day 20 (1X-2X). (C) Representative traces of evoked CM9 EPSPs and spontaneous miniature EPSPs (CM9 mEPSPs) from electrophysiological recordings of CM9 muscle fibers from flies subjected to the indicated dietary conditions. Scale = 1 mv, 10 ms. (D–F) Graphs represents the mean values for evoked EPSPs (D), mEPSPs (E), and quantal content (F) determined from recordings of CM9 muscle fibers from flies subjected to the indicated dietary condition or subjected to a diet shift (1X-2X) for 12 or 24 hr. Error bars = s.e.m. *p<0.05 determined using ANOVA. (G) Example traces of evoked EPSPs from paired-pulse experiments utilizing an inter-pulse interval (IPI) of 50 ms. (H) Graph represent the mean percent depression at indicated IPI. Error bars = s.e.m. **p<0.01, Student's t-test. (I) Representative traces of electrophysiological recordings of hyperosmotic-induced spontaneous release events from CM9 NMJs in animals raised for 21 days on indicated diet conditions incubated in hyperosmotic recording saline. Inserts represent broader timescale of boxed regions from traces. Scale = 1 s. (J) Histogram representing the spontaneous event frequency and K, the average number of total spontaneous release events observed during 1 min of hyperosmotic recordings.

The following source data is available for figure 1:

**Source data 1.** File contains the values represent the average value for the spontaneous release events per second determined in 5 s increments during the hypertonic stimulation of synaptic vesicle fusion at CM9 NMJs in animals raised on a 1X or 2X diet presented in *Figure 1J*.

on food containing 100 mg/ml of yeast (1X) release nearly twice as much neurotransmitter, represented as quantal content (the number of quanta per action potential) (*Fatt and Katz, 1951*), compared to flies raised on food containing 200 mg/ml of yeast (2X) (*Figure 1B–F*; see *Table 1* for all electrophysiological recording data). We further find that shifting 20-day-old flies from the 1X diet to the 2X diet (*Figure 1B*) resulted in a gradual reduction in the quantal content (QC) that reaches the level of neurotransmission observed in 2X animals within 24 hr of diet shift (*Figure 1C and F*) supporting that the effects of high-protein diet on release are likely not due to the accumulation of diet-related pathologies that result in reduced neurotransmitter release. In these recordings, we

**Table 1.** Quantal analysis of neurotransmission at the CM9 NMJ.

| Genotype (condition) | Diet | N | mEPSP (mV) | EPSP (mV) | QC | RMP (mV) | IR (MΩ) |
|---|---|---|---|---|---|---|---|
| $w^{1118}$ | 1X | 8 | 0.94 ± 0.04 | 3.46 ± 0.30 | 3.66 ± 0.28 | −40.89 ± 1.37 | 7.56 ± 0.80 |
| $w^{1118}$ | 2X | 8 | 0.83 ± 0.04 | 1.65 ± 0.08 | 2.01 ± 0.12 | −39.67 ± 0.57 | 7.00 ± 0.80 |
| $w^{1118}$ (12 hr shift) | 1-2X | 8 | 0.88 ± 0.02 | 2.74 ± 0.21 | 3.12 ± 0.24 | −38.40 ± 2.17 | 8.13 ± 1.01 |
| $w^{1118}$ (24-hr shift) | 1-2X | 8 | 0.89 ± 0.03 | 2.25 ± 0.27 | 2.58 ± 0.27 | −35.65 ± 1.53 | 8.75 ± 0.62 |
| E49-Gal4/+ | 1X | 8 | 0.96 ± 0.03 | 3.40 ± 0.16 | 3.55 ± 0.16 | −35.53 ± 3.24 | 7.48 ± 0.55 |
| E49-Gal4/+ | 2X | 8 | 0.92 ± 0.02 | 2.06 ± 0.09 | 2.23 ± 0.07 | −32.24 ± 0.83 | 7.12 ± 0.58 |
| UAS-4eBP$^{RNAi}$/+ | 1X | 8 | 0.94 ± 0.04 | 3.31 ± 0.33 | 3.50 ± 0.30 | −41.02 ± 1.40 | 7.88 ± 0.79 |
| UAS-4eBP$^{RNAi}$/+ | 2X | 8 | 0.83 ± 0.02 | 1.62 ± 0.12 | 1.98 ± 0.17 | −39.67 ± 0.57 | 8.25 ± 0.82 |
| UAS-4eBP$^{RNAi}$/+ | 1-2X | 8 | 0.93 ± 0.01 | 1.90 ± 0.05 | 2.05 ± 0.07 | −37.96 ± 0.54 | 8.25 ± 0.62 |
| E49-Gal4/+; UAS-4eBP$^{RNAi}$/+ | 1X | 7 | 0.92 ± 0.02 | 1.50 ± 0.06 | 1.64 ± 0.07 | −38.88 ± 0.68 | 9.14 ± 0.77 |
| E49-Gal4/+; UAS-4eBP$^{RNAi}$/+ | 2X | 8 | 0.91 ± 0.03 | 1.67 ± 0.13 | 1.86 ± 0.19 | −38.13 ± 0.50 | 8.00 ± 1.00 |
| E49-Gal4/+; UAS-4eBP$^{RNAi}$/+ | 1-2X | 8 | 0.94 ± 0.02 | 1.79 ± 0.10 | 1.90 ± 0.11 | −38.68 ± 0.67 | 8.75 ± 0.62 |
| UAS-chico$^{RNAi}$/+ | 1X | 8 | 0.90 ± 0.04 | 3.56 ± 0.33 | 3.96 ± 0.30 | −37.31 ± 1.49 | 8.63 ± 0.30 |
| UAS-chico$^{RNAi}$/+ | 2X | 8 | 0.83 ± 0.02 | 1.74 ± 0.10 | 2.10 ± 0.15 | −39.45 ± 0.47 | 8.25 ± 0.73 |
| E49-Gal4/UAS-chico$^{RNAi}$ | 1X | 8 | 0.92 ± 0.03 | 4.60 ± 0.30 | 5.09 ± 0.43 | −34.75 ± 1.07 | 8.31 ± 0.47 |
| E49-Gal4/UAS-chico$^{RNAi}$ | 2X | 8 | 0.95 ± 0.05 | 4.09 ± 0.28 | 4.34 ± 0.24 | −38.52 ± 4.49 | 8.40 ± 0.77 |
| E49-Gal4/UAS-chico$^{RNAi}$ | 1-2X | 8 | 0.91 ± 0.04 | 4.02 ± 0.23 | 4.48 ± 0.28 | −35.47 ± 2.82 | 8.75 ± 0.68 |
| E49-Gal4/UAS-chico$^{RNAi}$; UAS-4eBP$^{RNAi}$/+ | 2X | 8 | 0.84 ± 0.05 | 2.18 ± 0.12 | 2.60 ± 0.13 | −39.51 ± 1.96 | 7.75 ± 0.85 |
| UAS-InR$^{DN}$/+ | 2X | 8 | 0.86 ± 0.01 | 2.07 ± 0.17 | 2.42 ± 0.21 | −32.19 ± 1.55 | 8.06 ± 0.79 |
| E49-Gal4/UAS-InR$^{DN}$ | 2X | 8 | 0.85 ± 0.03 | 2.87 ± 0.15 | 3.43 ± 0.24 | −35.92 ± 2.20 | 8.69 ± 0.54 |
| $w^{1118}$ | 1X | 8 | 0.92 ± 0.02 | 3.25 ± 0.25 | 3.53 ± 0.26 | −34.61 ± 1.77 | 7.88 ± 0.69 |
| $w^{1118}$ | 2X | 8 | 0.84 ± 0.03 | 1.82 ± 0.10 | 2.18 ± 0.14 | −36.62 ± 1.14 | 8.50 ± 0.80 |
| $w^{1118}$ (+CXM) | 1-2X | 8 | 0.99 ± 0.04 | 4.31 ± 0.20 | 4.39 ± 0.26 | −40.58 ± 1.84 | 7.88 ± 0.69 |
| $w^{1118}$ (+Veh (CMX)) | 1-2X | 8 | 0.95 ± 0.02 | 2.50 ± 0.11 | 2.63 ± 0.11 | −40.01 ± 2.56 | 8.00 ± 0.68 |
| $w^{1118}$ (+CXM) | 1X | 8 | 1.04 ± 0.03 | 4.23 ± 0.23 | 4.09 ± 0.25 | −39.09 ± 0.89 | 9.00 ± 0.82 |
| $w^{1118}$ (+rapamycin) | 1-2X | 8 | 0.86 ± 0.03 | 2.11 ± 0.13 | 2.48 ± 0.22 | −41.29 ± 1.19 | 7.63 ± 0.78 |
| $w^{1118}$ (+Veh (rapa)) | 1-2X | 8 | 0.82 ± 0.03 | 2.05 ± 0.20 | 2.54 ± 0.29 | −39.87 ± 1.84 | 6.88 ± 0.61 |
| $w^{1118}$ | 1X | 8 | 0.89 ± 0.03 | 3.37 ± 0.20 | 3.81 ± 0.27 | −31.25 ± 1.47 | 8.25 ± 0.75 |
| $w^{1118}$ | 2X | 5 | 0.95 ± 0.03 | 1.91 ± 0.16 | 2.00 ± 0.13 | −34.05 ± 1.48 | 7.80 ± 0.97 |
| dFOXO$^{del94}$/dFOXO$^{21}$ | 1X | 8 | 0.94 ± 0.02 | 1.98 ± 0.13 | 2.10 ± 0.13 | −34.60 ± 1.35 | 8.44 ± 0.48 |
| dFOXO$^{del94}$/dFOXO$^{21}$ | 2X | 8 | 0.96 ± 0.03 | 1.69 ± 0.09 | 1.77 ± 0.13 | −38.21 ± 1.52 | 7.75 ± 0.75 |
| dFOXO$^{del94}$/dFOXO$^{21}$ | 1-2X | 8 | 0.94 ± 0.01 | 1.58 ± 0.04 | 1.68 ± 0.03 | −34.54 ± 2.06 | 8.69 ± 0.74 |
| dFOXO$^{del94}$ / dFOXO$^{21}$ , UAS-4eBP | 1X | 8 | 0.94 ± 0.04 | 1.95 ± 0.23 | 2.06 ± 0.19 | −31.36 ± 2.83 | 7.38 ± 0.74 |
| E49-Gal4/+; dFOXO$^{del94}$ / dFOXO$^{21}$ , UAS-4eBP | 1X | 8 | 0.88 ± 0.04 | 3.28 ± 0.22 | 3.85 ± 0.43 | −31.83 ± 2.65 | 6.88 ± 0.75 |
| UAS-4eBP/+ | 1X | 8 | 0.96 ± 0.04 | 3.30 ± 0.16 | 3.47 ± 0.16 | −33.08 ± 1.03 | 7.79 ± 0.38 |
| E49-Gal4/+;UAS-4eBP/+ | 1X | 8 | 0.92 ± 0.03 | 4.79 ± 0.38 | 5.25 ± 0.46 | −34.77 ± 2.12 | 8.08 ± 0.58 |
| E49-Gal4/UAS-stauen$^{RNAi}$ | 1X | 8 | 0.89 ± 0.03 | 3.37 ± 0.20 | 3.81 ± 0.27 | −36.32 ± 1.22 | 8.32 ± 0.66 |
| E49-Gal4/UAS-staufen$^{RNAi}$ | 1-2X | 8 | 0.95 ± 0.04 | 3.45 ± 0.21 | 3.65 ± 0.16 | −39.64 ± 2.42 | 7.55 ± 0.32 |
| +/UAS-staufen$^{RNAi}$ | 1-2X | 9 | 0.90 ± 0.04 | 2.35 ± 0.13 | 2.68 ± 0.20 | −35.51 ± 1.21 | 8.02 ± 0.73 |
| $W^{1118}$ | 1X | 8 | 0.93 ± 0.02 | 3.21 ± 0.19 | 3.47 ± 0.22 | −30.56 ± 1.49 | 8.31 ± 0.09 |
| $W^{1118}$ | 2X | 8 | 0.95 ± 0.02 | 1.96 ± 0.15 | 2.07 ± 0.17 | −30.43 ± 1.20 | 8.06 ± 0.67 |

*Table 1 continued on next page*

Mahoney *et al*. eLife 2016;5:e16807. DOI: 10.7554/eLife.16807

*Table 1 continued*

| Genotype (condition) | Diet | N | mEPSP (mV) | EPSP (mV) | QC | RMP (mV) | IR (MΩ) |
|---|---|---|---|---|---|---|---|
| +/+,cpx[SH1]/+ | 1X | 8 | 0.91 ± 0.01 | 4.23 ± 0.48 | 4.66 ± 0.51 | −32.32 ± 1.40 | 7.88 ± 0.74 |
| +/+,cpx[SH1]/+ | 2X | 8 | 0.96 ± 0.01 | 2.65 ± 0.30 | 2.75 ± 0.32 | −31.26 ± 2.87 | 7.94 ± 0.83 |
| UAS-Complexin/+ | 1X | 9 | 0.99 ± 0.05 | 4.15 ± 0.46 | 4.37 ± 0.56 | −36.12 ± 1.65 | 6.67 ± 0.67 |
| UAS-Complexin/+ | 2X | 9 | 0.95 ± 0.02 | 2.39 ± 0.19 | 2.54 ± 0.24 | −31.27 ± 1.99 | 7.22 ± 0.80 |
| E49-Gal4/UAS-Complexin | 1X | 9 | 0.90 ± 0.05 | 2.22 ± 0.28 | 2.48 ± 0.28 | −30.59 ± 1.97 | 6.79 ± 0.73 |
| E49-Gal4/UAS-Complexin | 2X | 9 | 0.98 ± 0.05 | 2.58 ± 0.21 | 2.63 ± 0.18 | −34.86 ± 2.42 | 7.72 ± 0.52 |

Table contents ordered by order of appearance in body of text. All values represent the average value ± sem (N = animals, 1 recording per animal). For each recording, the EPSP value represents the average of 60 evoked responses and the value for mEPSP represents the average of 30 events. All stocks were backcrossed five generations and re-established in the $w^{1118}$ background. Quantal content (QC) is determined for each NMJ by dividing the amplitude of the EPSP by the amplitude of the mEPSP for each recording. RMP = resting membrane potential of CM9 muscle fiber. IR = depolarizing input resistance of CM9 muscle fiber.

observed no effect of diet on the amplitude of the spontaneous release events (mEPSPs), the resting membrane potential, or the resistance of the muscle demonstrating that the effects of diet on pre-synaptic function are not due to changes in the excitability of the post-synaptic CM9 muscle fibers (*Figure 1E*; *Table 1*) (*Davis, 2013*). Paired pulse analysis at the CM9 NMJ reveals that flies raised on 1X diet show pronounced synaptic depression when EPSPs were evoked with a 50-ms interpulse interval that was absent at CM9 NMJs in flies raised on the 2X diet (*Figure 1G and H*). Under our recording conditions, this result is consistent with a reduction in the probability of release at CM9 NMJs in animals subjected to the 2X diet (*Zucker and Regehr, 2002*). Hypertonic challenge of NMJs with sucrose has been used to estimate the size of the readily-releasable pool at Drosophila NMJs (*Mahoney et al., 2014*; *Müller and Davis, 2012*; *Yoshihara et al., 2010*). We find that there is no significant difference in the size of the sucrose-sensitive pool of synaptic vesicles (SVs) at the CM9 NMJs in flies raised on a 1X diet compared to flies raised on a 2X diet (*Figure 1I–K*) (*Rosenmund and Stevens, 1996*). Combined with our previous observations of a lack of effect of diet on synaptic area (*Rawson et al., 2012*), these data support that increases in the protein content of the diet reduces the probability of SV release at the CM9 NMJ.

## Cell autonomous insulin signaling regulates neurotransmission via 4eBP

To determine what signaling systems within the motor neuron are responsible for the effects of diet on neurotransmission, we used motor-neuron-specific RNAi to screen important nutrient-sensing pathways using viability in a diet sensitive *glued* mutant fly background (*Rawson et al., 2012*) followed by analysis of promising candidates using the proboscis extension reflex (PER) (*Figure 2A*) (*Gordon and Scott, 2009*; *Kimura et al., 1986*). This motor reflex requires the CM9 motor neuron and provides a simple assay for investigating CM9 motor neuron function by analyzing the velocity of proboscis extension using particle-tracking software to track bristle paths during the PER (*Figure 2A*, panels i-iv) (*Rawson et al., 2012*). For these analyses, we combine the *E49-Gal4* driver, which is expressed in a few number of neurons in the adult including the CM9 motor neuron, with gene-specific RNAi allowing us to focus on the cell autonomous effects of insulin signaling without grossly altering whole animal insulin signaling (*Gordon and Scott, 2009*; *Rawson et al., 2012*). This approach identified *thor*, the *Drosophila* homologue of eukaryotic initiation factor 4e binding protein (4eBP), as a critical presynaptic mediator of the positive effects of the 1X diet on motor function (*Figure 2B and C*). Analysis of neurotransmission in these animals found that knockdown of *4eBP* in the CM9 motor neuron reduced the presynaptic release of neurotransmitter in animals raised on a 1X diet compared to controls (*Figure 2F–I*; see *Table 1* for *E49-Gal4/+* control values). We also observe no difference between neurotransmitter release in *4eBP* knockdown animals (*4eBP[RNAi]*) raised on 1X, 2X, or 1-2X diet conditions consistent with the effects of 4eBP knockdown being specific to diet regulation of neurotransmission and not basal release (*Figure 2F–I*). The effectiveness and specificity of all RNAi constructs were determined using quantitative RT-PCR and finds that the

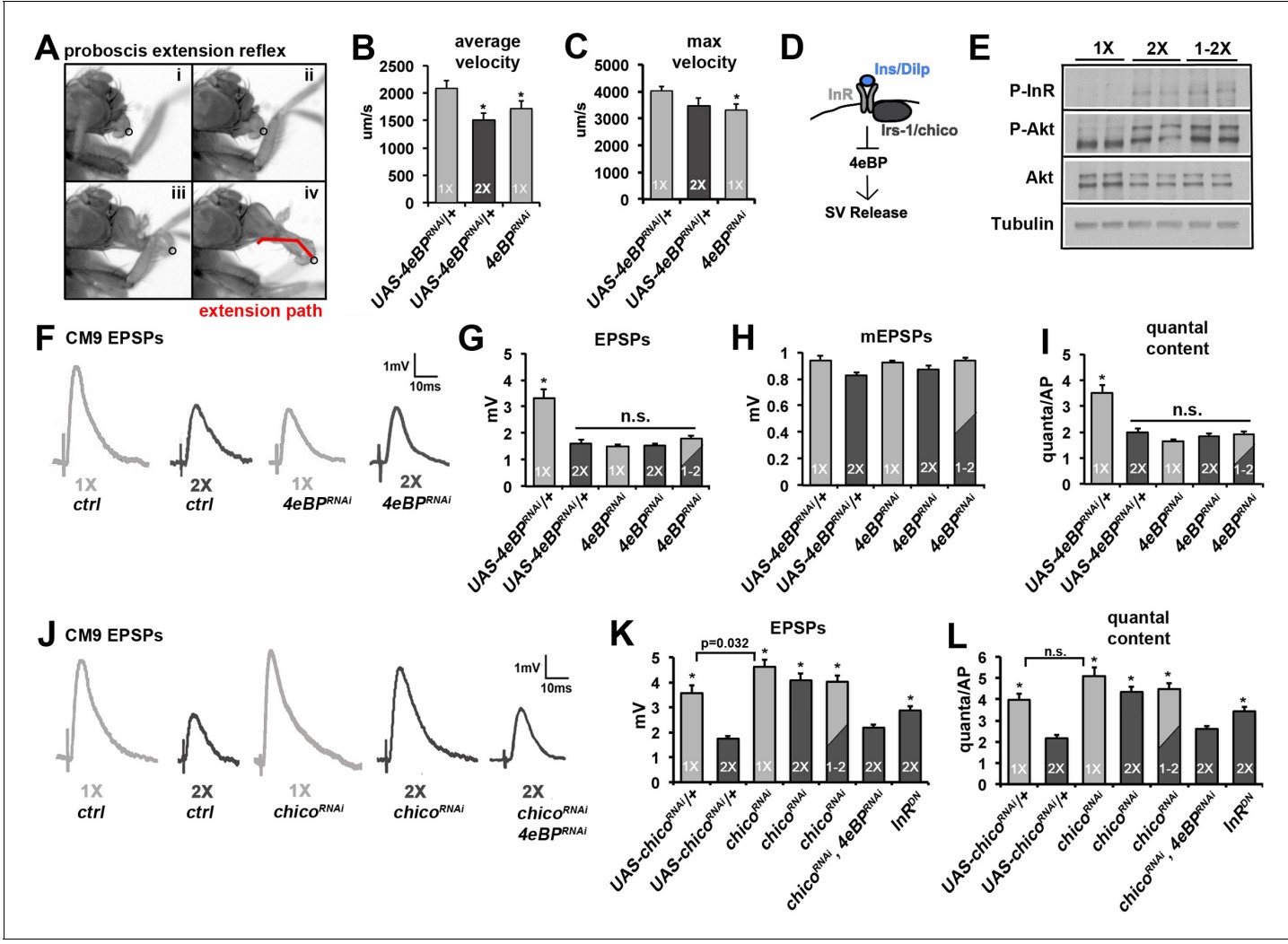

**Figure 2.** Insulin/DILP signaling negatively regulates presynaptic release at the CM9 NMJ. (**A**) Images from a proboscis extension reflex (PER) in response to tarsal stimulation with 0.5 M sucrose. Circle indicates location of sensory bristles tracked during the extension event resulting in an extension path (red line in panel iv) that is used for analysis of velocity. (**B** and **C**) Graphs represent the mean values for (**B**) average velocity and (**C**) max velocity for indicated genotypes and dietary conditions. All RNAi knock-downs utilize the Gal4:UAS binary expression system by combining the transgenic *UAS* construct (i.e. *UAS-4eBP^RNAi*) with the CM9 motor neuron-specific *E49-Gal4* driver. *p<0.05 versus 1X controls determined using ANOVA. Error bars = s.e.m. (**D**) Pathway represents the putative effects of insulin signaling on SV exocytosis. (**E**, **F**) CM9 EPSP traces of indicated genotype and dietary condition demonstrating the requirement for 4eBP on neurotransmission in animals raised on the 1X and 2X diets. Scale bar = 1 mV, 10 ms. (**G**–**I**) Graphs represent the average values for CM9 EPSPs (**G**), mEPSPs (**H**), and quantal content (**I**) determined from CM9 recordings from 21-day-old flies of indicated genotypes raised on indicated dietary conditions. Error bars = s.e.m. * indicates values significantly different from all other values determined using ANOVA (p<0.01). (**J**) CM9 EPSP traces of indicated genotype and dietary condition demonstrating the requirement for *chico* on neurotransmission in animals raised on the 1X and 2X diets. The effect of *chico* knock-down (*chico^RNAi*) on neurotransmission in animals raised on a 2X diet is suppressed by knockdown of *4eBP* consistent with 4eBP functioning downstream of Chico. Scale bar = 1 mV, 10 ms. (**K** and **L**) Graphs represent the average values of EPSPs (**K**) and quantal content (**L**) determined from CM9 recordings from 21-day old flies of indicated genotypes raised on indicated dietary conditions. Error bars = s.e.m. *p<0.05 versus 2X controls determined using ANOVA.

The following figure supplements are available for figure 2:

**Figure supplement 1.** Effects of diet and neuronal insulin signaling on SV exocytosis at larval NMJ.

**Figure supplement 2.** Cycloheximide blocks the effects of diet switch on SV exocytosis.

reductions in *4eBP* mRNA levels are approximately 60% in control experiments (data not shown). All values for the electrophysiological analyses are listed in *Table 1*.

The activity of 4eBP is negatively regulated by the insulin signaling system (*Figure 2D*) (*Laplante and Sabatini, 2012*), a signaling system critical for integrating the nutritional status of the organism with cellular metabolism and organ function. Previous studies in *Drosophila* using similar dietary conditions have shown that changes in diet can alter insulin signaling (*Grönke et al., 2010*; *Morris et al., 2012*). To confirm that our diet conditions resulted in changes in insulin signaling, we performed immunoblot analysis of both phosphorylated Insulin receptor (InR) and phosphorylated Akt, reporters of increased insulin signaling (*Figure 2E*) (*Bai et al., 2015*; *Bjedov et al., 2010*). This analysis found that both the 2X diet and the 1-2X diet shift conditions resulted in increased phosphorylation of InR and Akt compared to the 1X diet condition supporting that our 2X and 1-2X diet conditions result in increased insulin signaling in our flies. Consistent with insulin signaling in the CM9 motor neuron being responsible for the effects of diet on SV release, we observe that CM9-specific knock-down of the Drosophila IRS-1 homologue *chico* (*chico^{RNAi}*) in flies raised on the 2X and 1-2X shift diets resulted in a significant increase SV release compared to the 2X controls (*Figure 2J–L*). This effect of *chico^{RNAi}* on SV release in flies raised on the 2X diet was phenocopied by the overexpression of a dominant negative insulin receptor (*InR^{DN}*) in the CM9 motor neuron (*Figure 2K and L*) (*Peru Y Colón de Portugal et al., 2012*). We also observe that there is no difference in neurotransmitter release between *chico^{RNAi}* animals raised on 1X, 2X or 1-2X diets versus 1X controls (*Figure 2K and L*), except that *chico^{RNAi}* animals raised on 1X have slightly increased EPSPs (p=0.032) compared to 1X controls consistent with low-level insulin signaling even in animals raised on a 1X diet. Finally, the increase in neurotransmitter release observed in the *chico^{RNAi}* flies raised on the 2X diet was suppressed by the simultaneous knock-down of *4eBP* (*Figure 2K–L*), consistent with 4eBP functioning downstream of Chico during the regulation of neurotransmission in response to diet. This epistasis between *chico* and *4eBP* is similar to what has been recently observed for *Drosophila* lifespan (*Bai et al., 2015*). The regulation of neurotransmission by insulin signaling appears to be specific to the adult life stage since we do not observe the same effects of diet or presynaptic knockdown of *4eBP* or *chico* on SV release from larval NMJs (*Figure 2—figure supplement 1*).

Because of the role of 4eBP in the inhibition of translation, our data suggests that insulin signaling results in the translation of a negative regulator(s) of SV release (*Figure 2—figure supplement 2A*). To investigate this model, flies were raised on a 1X diet were fed the protein translation inhibitor cycloheximide (cmx) for 1 hr prior to being shifted to 2X diet supplemented with cmx for 24 hr (*Figure 2—figure supplement 2B*). We predict that this 1-2X shift diet results in an increase in insulin signaling within the CM9 motor neuron resulting in increased protein translation, which is supported by our immunoblot analysis (*Figure 2F*). We find that cycloheximide effectively inhibits the reduction in SV release in response to a shift from 1X to 2X diet conditions (*Figure 2—figure supplement 2C and E*) without significant effects on the amplitudes of the mEPSPs (*Figure 2—figure supplement 2D*). Taken together, these data are consistent with increased insulin signaling resulting in the translation of a negative regulator(s) of SV release.

## The control of neurotransmission by insulin is FOXO-dependent

The activity of *Drosophila* 4eBP can be positively regulated transcriptionally by the *Drosophila* forkhead transcription factor dFOXO (*Figure 3A*) (*Puig et al., 2003*; *Teleman et al., 2005*). Analysis of *4eBP* mRNA levels in thoracic motor neurons purified by FACS from flies raised on a 1X diet, a 2X diet, or subjected to a diet switch from a 1X diet to a 2X diet reveals that *4eBP* mRNA levels are sensitive to diet and that during diet shift the declines in *4eBP* mRNA levels (*Figure 3B*) correlate with our observed declines in SV release (*Figure 1F*). This suggested that the effects of diet on SV release are due to the transcriptional regulation of *4eBP*. Previous studies have identified dFOXO-binding sites near the 5′ end of the *4eBP* gene (*Puig et al., 2003*). Using chromatin immunoprecipitation (ChIP) with anti-FOXO antibodies, we found that these dFOXO-binding sites in the *4eBP* promoter region were enriched in our dFOXO ChIP of thoracic ganglion isolated from animals raised on a 1X diet as compared to animals raised on a 2X diet (*Figure 3C*). This difference was not due to changes in *dFOXO* protein levels under our diet conditions (*Figure 3D*). This supports that dFOXO binding to the *4eBP* gene is increased under our 1X diet condition compared to the 2X diet condition consistent with dFOXO driving the expression of *4eBP* under 1X diet conditions. Importantly, these molecular data suggest that the regulation of 4eBP mRNA levels by diet is conserved among all motor

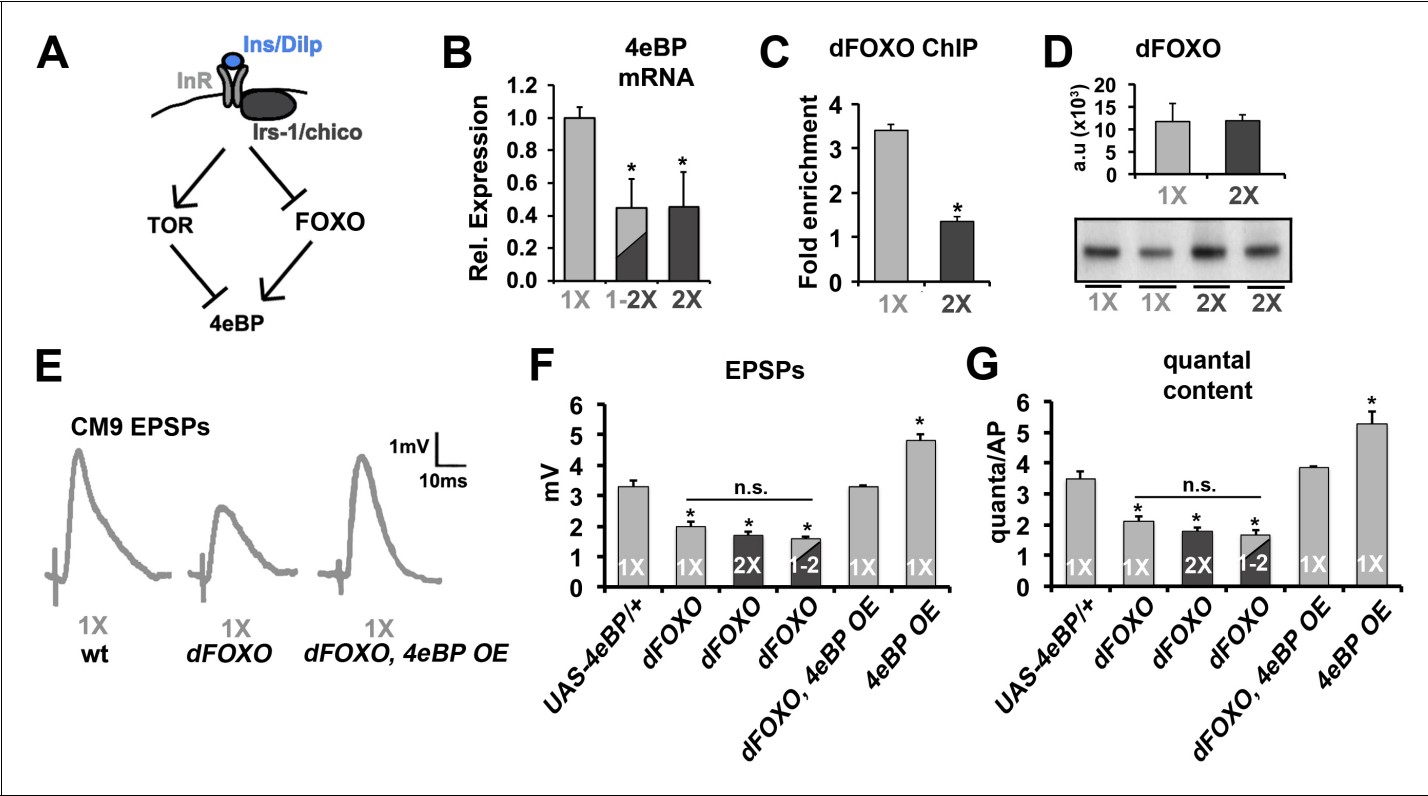

**Figure 3.** Effects of diet on the release of neurotransmitter requires FOXO. (A) Diagram depicts the regulation of 4eBP by either FOXO-dependent transcription or dTOR-dependent phosphorylation. (B) Relative mRNA expression levels of *4eBP* in purified motor neurons from 21-day-old animals raised on the indicated diet conditions. (C) Graphs represent the average relative fold enrichment of 4eBP DNA in anti-dFOXO chromatin immunoprecipitations (ChIPs) from thoracic ganglions isolated from animals raised on 1X or 2X diets. (D) Graphs represent average dFOXO protein levels estimated from flies used for ChIP. Values were normalized to actin. Error bars = s.e.m. Immunoblot of dFOXO is shown below. (E) Representative CM9 EPSP traces from 14-day-old flies raised on 1X diet of the indicated genotypes. In these genotypes, the overexpression of 4eBP is restricted to the CM9 MN using the *E49-Gal4* driver. (F and G) Graphs represent the mean value for EPSPs (F) and quantal content (G) for indicated genotypes raised for 14 days on indicated diets. Error bars = s.e.m. *p<0.05 versus 1X wild-type controls determined using ANOVA.

neurons and not specific to CM9 MNs. Electrophysiological recordings from CM9 NMJs in *dFOXO* mutants raised on 1X, 2X or 1-2X shift diets found that neurotransmission is reduced in *dFOXO* mutants compared to 1X controls but are not different than the 2X controls, similar to what we observed in the *4eBP^RNAi* flies (*Figure 3E–G*, *Table 1*). Further, this deficit in neurotransmission at the CM9 NMJs is reversed by the over-expression of *4eBP* in the CM9 motor neuron in *dFOXO* mutants (*Figure 3E–G*). We also observe that overexpression of *4eBP (4eBP OE)* increases neurotransmission compared to 1X controls consistent with persistent insulin signaling in animals on the 1X diet (*Figure 3F and G*). These results support the model that the negative regulation of neurotransmitter release by insulin signaling involves repression of the *dFOXO*-dependent gene transcription of the *4eBP* locus.

The phosphorylation, and subsequent inhibition, of 4eBP by mTOR is an established mechanism for regulating protein translation in response to changes in diet (*Figure 4A*) (*Gingras et al., 1999*) (*Ma and Blenis, 2009*). Previous studies have established that postsynaptic TOR signaling can influence synapse function (*Penney et al., 2012*; *Weston et al., 2012*). Furthermore, TOR signaling has been linked to a number of important neuronal processes including the regulation of synapse structure and function (*Bidinosti et al., 2010*; *Costa-Mattioli et al., 2009*; *Kelleher et al., 2004*; *Stoica et al., 2011*). Thus, we wanted to investigate if *Drosophila* TOR signaling also played a role in the effects of diet on SV release. To test this possibility, flies raised for 14 days on 1X diet were placed on 1X food supplemented with the potent TOR inhibitor Rapamycin for 6 days prior to switching to a 2X diet also supplemented with Rapamycin (*Figure 4B*). This treatment paradigm was

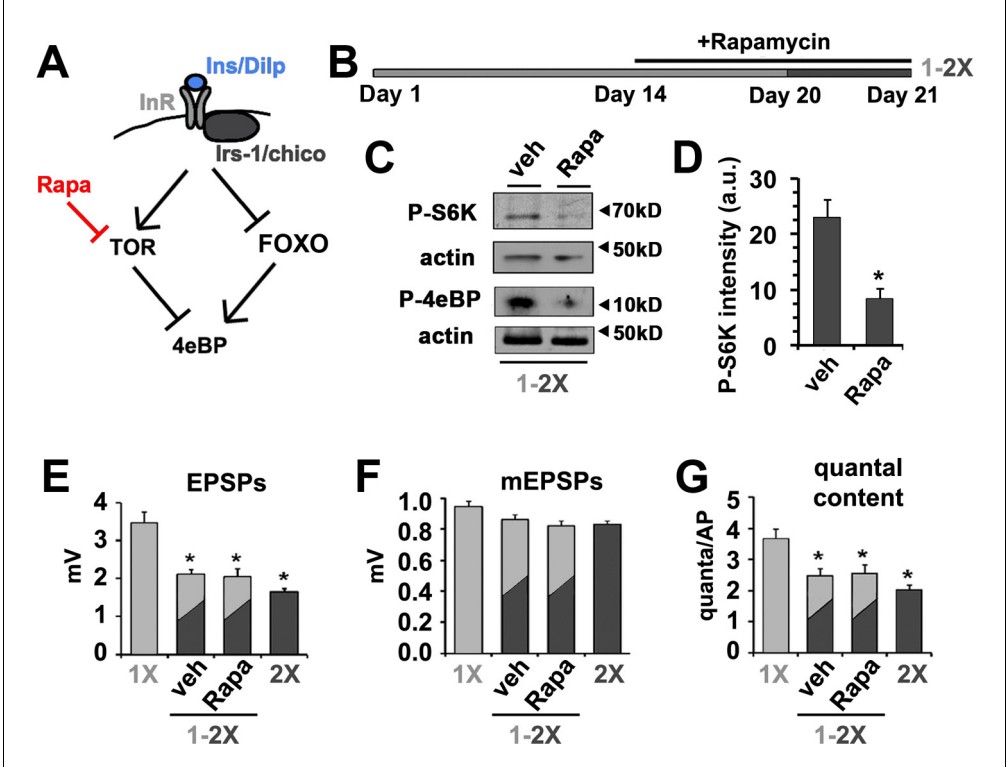

**Figure 4.** Effects of diet on the release of neurotransmitter is independent of dTOR. (**A**) Diagram depicts the regulation of 4eBP by either FOXO-dependent transcription or dTOR-dependent phosphorylation indicating the effects of rapamycin. (**B**) To investigate the effect of rapamycin (Rapa) on diet-regulated SV exocytosis, animals were fed for 14 days on 1X food and then switched to a 1X food supplemented with either 200 µM rapamycin or vehicle for 6 more days. On day 20, animals were switched from a 1X to a 2X diet supplemented with rapamycin or vehicle for 24 hr prior to electrophysiological analyses. (**C**) Immunoblots of phosphorylated S6 kinase (P-S6K) or 4eBP (P-4eBP) from animals subjected to above rapamycin treatment demonstrating effective inhibition of dTOR kinase activity under these dietary conditions. Actin signals serves as protein loading control. (**D**) Quantification of intensity of P-S6K determined from immunoblots and normalized for loading. *p<0.05 determined using Student's T-test. (**E**–**G**) Graphs represent the mean values for EPSPs (**E**), mEPSP (**F**) and quantal content (**G**) recorded from CM9 NMJs of 21-day-old wild-type flies of indicated dietary condition. Error bars = s.e.m. *p<0.05 versus 1X controls determined using ANOVA.

sufficient to reduce the phosphorylation of S6 kinase (P-S6K) and 4eBP (P-4eBP) supporting successful inhibition of dTOR under these feeding paradigm (*Figure 4C and D*). Despite the change in phosphorylation of 4eBP, we observed no effect of the rapamycin treatment on the reduction of neurotransmitter release observed in response to the 1X to 2X diet shift (*Figure 4E–G*). These data are consistent with the effects of insulin signaling on SV exocytosis being largely independent of dTOR signaling.

## Complexin is a target of insulin signaling in CM9 motor neurons

Our data suggest that ultimately insulin signaling was controlling the translation of an existing mRNA(s) that altered neurotransmitter release. An analogous process is the translational control of post-synaptic function by BDNF, which utilizes RNA particles consisting of target mRNAs, translational machinery, transport proteins and RNA-binding proteins such as the Staufen-family of RNA-binding proteins (*Leal et al., 2014*; *Takei et al., 2004*). Therefore, we investigated whether Drosophila Staufen was involved in the effects of diet on neurotransmitter release at the CM9 NMJ. Similar to what we observed with cycloheximide, we found the RNAi knockdown of Staufen (*staufen^RNAi*) in CM9 motor neurons blocked the reduction in neurotransmitter in response to diet shift

(*Figure 5A,C and D*) without altering muscle sensitivity to glutamate (*Figure 5B*; *Table 1*). These results support the model that a Staufen bound mRNA is involved in the effects of diet on neurotransmitter release. Recently, the collection of mRNAs bound to Staufen2 in mammalian brain tissues was defined and a number of important presynaptic regulators of synapse function were identified (*Heraud-Farlow et al., 2013*). One of the mRNAs identified in this study was the mRNA encoding *complexin,* a small peptide that functions as both a facilitator and an inhibitor of SV exocytosis (*Südhof, 2012*; *Trimbuch and Rosenmund, 2016*). In addition, *Drosophila complexin* mRNA contains predicted Staufen target sequences (STSs) (*Laver et al., 2013*). Therefore, we investigated if *complexin* mRNA was bound to Staufen in motor neurons in adult Drosophila. FACs sorted motor neurons expressing a GFP-tagged Staufen were subjected to an RNA immunoprecipitation (RIP) with anti-GFP-coated beads (*Laver et al., 2013*). Enrichment of *complexin* mRNA in anti-GFP RIPs from experimental versus control motor neurons was determined using quantitative RT-PCR. These analyses revealed that *complexin* mRNA was highly enriched in Staufen RIPs from motor neurons (*Figure 5E*). Surprisingly, we also observed that *4eBP* mRNA was also highly enriched even though 4eBP mRNAs do not contain a predicted STS (*Figure 5E*). The RIP of *complexin* and *4eBP* mRNA was specific since a number of other neuronal mRNAs were not enriched in these RIP experiments including mRNAs for *tubulin, syntaxin* and the *cacophony* voltage-gated calcium channel (*Figure 5E*).

Based on these results, we investigated if synaptic Complexin levels are sensitive to diet conditions. Immunofluorescent microscopy of CM9 NMJs in flies raised on 1X and 2X diets found that Complexin staining was increased at CM9 NMJs in animals on a 2X diet compared to animals on the 1X diet (*Figure 6A and B*). We used deconvolution microscopy to quantify the intensity of Complexin staining at the CM9 NMJ and find that the average maximum pixel intensity of Complexin staining is significantly increased at NMJs from animals raised on the 2X diet compared to the 1X diet (*Figure 6C*) and that the distribution of m.p.i values is also significantly different (*Figure 6D*; see source data file for Kolmogorov-Smirmov test results). Furthermore, *chico^RNAi* animals raised on a 2X diet had significantly reduced Complexin levels (*Figure 6C*). Identical results were observed for median and mean pixel intensity as well (data not shown). Because our molecular analyses suggested that this signaling system also functioned in motor neurons located within the thoracic ganglion, we investigate synaptic Complexin levels at the NMJs formed on the lateral abdominal muscles (LAMs), which are innervated by motor neurons found within the thoracic ganglion (*Hebbar et al., 2006*; *Krzemien et al., 2012*). Using an identical approach to quantifying Complexin at the CM9 NMJs, we find that there is significantly more Complexin at the LAM NMJs in flies raised on a 2X diet

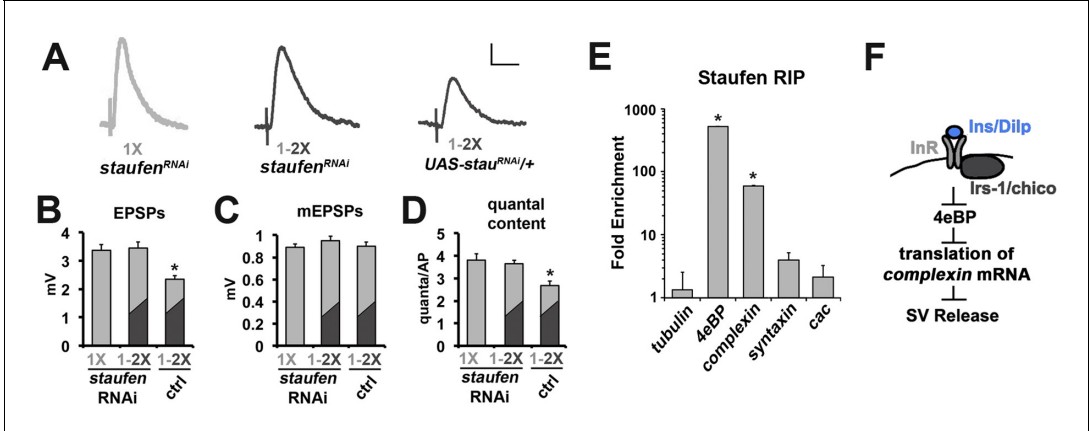

**Figure 5.** The role of Staufen during the regulation of neurotransmission by diet. (**A**) Representative traces of EPSPs from CM9 NMJs from 21 day old *staufen^RNAi* raised on a 1X diet or *staufen^RNAi* and control flies subjected to a diet switch from 1X to 2X diet on day 20 and recorded on day 21. Scale bar = 1 mv, 10 ms. (**B–D**) Graphs represent the mean values for EPSPs (**B**), mEPSPs (**C**), and quantal content (**D**) recorded from indicated genotypes. *p<0.05 determined using ANOVA. (**E**) Graphs represent the fold enrichment of Staufen-bound mRNAs immunoprecipitated from FACS sorted motor neurons. *p<0.01 versus tubulin control determined using ANOVA. (**F**) Diagram depicts the putative regulation of *complexin* mRNA translation in response to insulin signaling in the CM9 motor neuron.

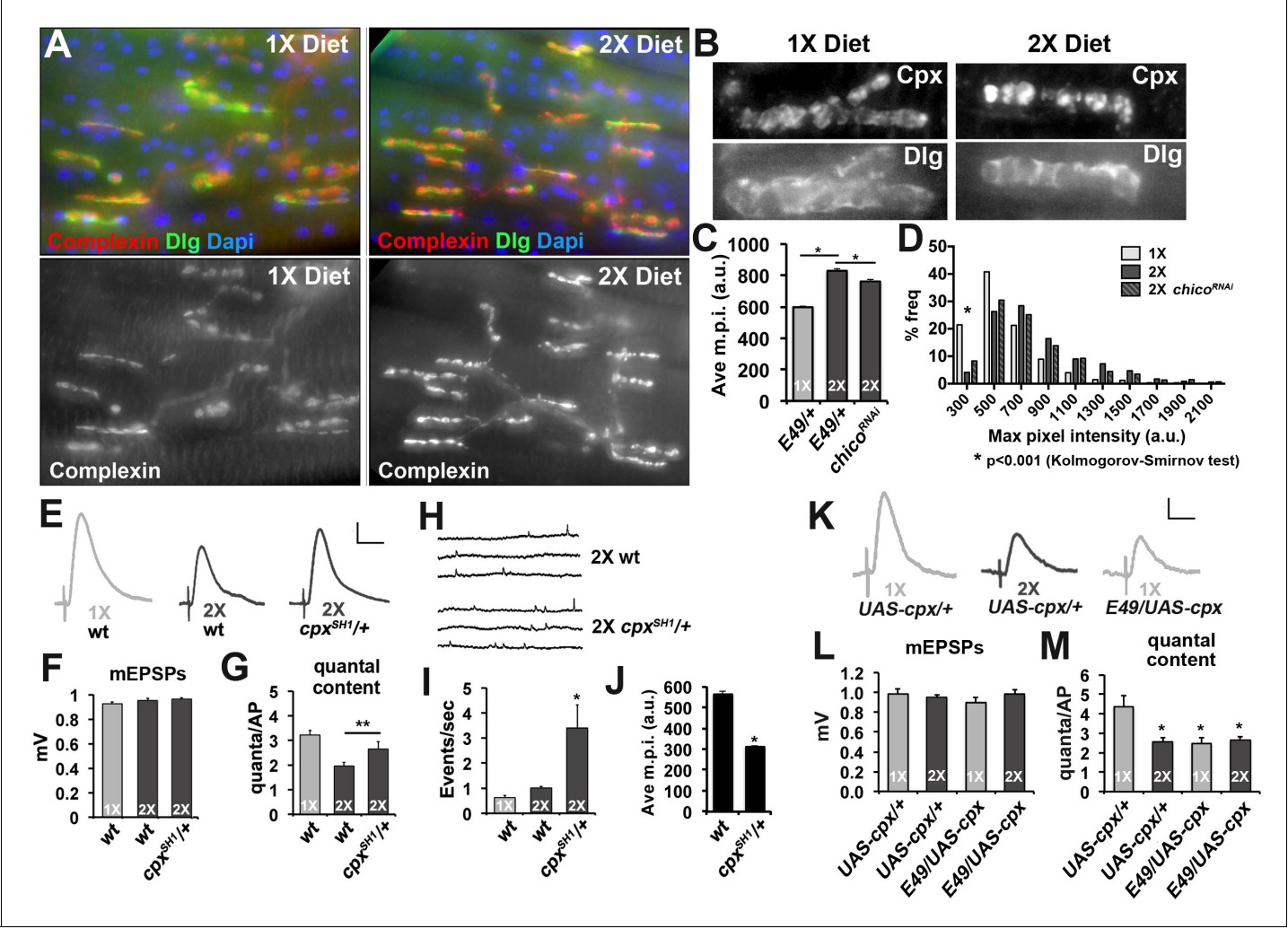

**Figure 6.** Complexin levels regulate SV release in response to diet. (**A**) Immunofluorescent images of CM9 NMJs from animals raised on a 1X (left panels) or 2X (right panels) diet co-stained for Complexin (red-upper panels and lower panels), Discs-large (Dlg, green upper panels) and Dapi (blue-upper panels). (**B**) High magnification of Complexin (Cpx-upper panels) and Discs-large (Dlg-lower panels) from CM9 NMJ boutons in animals raised on a 1X (left panels) or 2X (right panels) diet conditions. Images for Cpx have been deconvolved. (**C** and **D**) Graphs represent the average value (**C**) and frequency histogram (**D**) for the maximum pixel intensity (m.p.i.) of synaptic Complexin from indicated diet conditions and genotype. *p<0.05 determined using ANOVA comparison of mean values (**C**) or using a Kolmogorov-Smirnov test (**D**). (**E**) Representative traces of EPSPs from CM9 NMJs from wild type (*wt*) or *complexin* heterozygotes (*cpx^SH1^/+*) raised for 21 days on the indicated diets. Scale bar = 1 mV, 10 ms. (**F** and **G**) Graphs represent the mean values for mEPSPs (**F**) and quantal content (**G**) from CM9 NMJs from 21-day-old animals of indicated genotypes raised on the indicated diet. *p<0.05 versus 1X condition determined using ANOVA. (**H**) Representative traces from *wt* and *cpx^SH1^/+* animals raised on 2X diet for 21 days. Graphs below traces represent the quantification of the events per second from CM9 NMJs of the indicated genotypes raised on the indicated diets. *p<0.05 versus 1X control determined using ANOVA. (**I**) Representative traces of EPSPs from CM9 NMJs in 21-day-old animals on the indicated genotypes raised on the indicated diet. Scale bar = 1 mV, 10 ms. (**J** and **K**) Graphs represent the mean values for mEPSPs (**J**) and quantal content (**K**) from CM9 NMJs from 21-day-old animals of indicated genotypes raised on the indicated diets. *p<0.05 versus 1X control determined using ANOVA.

The following source data and figure supplement are available for figure 6:

**Source data 1.** File contains background-corrected values of max pixel intensity from complexin (Cpx) staining at the CM9 NMJ from indicated genotypes and diet conditions.

**Figure supplement 1.** Diet effects on synaptic complexin levels at the lateral abdominal muscle NMJs.

compared to flies raised on a 1X diet supporting that this signaling system is present in LAM motor neurons as well (*Figure 6—figure supplement 1B–D*). Functionally, we find that reduction of one copy of the *complexin* gene ($cpx^{SH1}$/+) significantly increases quantal content at CM9 NMJs in animals raised on both the 1X and 2X diets compared to controls (*Figure 6E and F*, *Table 1*). We also observe a significant increase in the frequency of spontaneous SV fusion events in $cpx^{SH1}$/+ animals consistent with previous observations at the *Drosophila* larval NMJ (*Figure 6H*) (*Jorquera et al., 2012*). Fluorescent microscopic quantification of synaptic complexin levels reveals a significant reduction in the amount of Complexin at the synapse in the $cpx^{SH1}$/+ animals compared to wild type (*Figure 6J*). Conversely, we observe a reduction in quantal content if we overexpress Complexin in the CM9 motor neuron of animals raised on a 1X diet compared to 1X diet controls (*Figure 6K–M*). We also find that overexpression of Complexin in animals on 2X diets does not reduce neurotransmission further compared to 2X diet controls similar to our data with *4eBP*, *chico* and *dFOXO* (*Figure 6L and M*). These data are consistent with the model that diet can influence neurotransmission at the adult NMJ by regulating the synaptic levels of Complexin.

## Discussion

Here, we have revealed that insulin signaling in adult *Drosophila* motor neurons can negatively regulate the release of neurotransmitter from the NMJ. This control of neurotransmission by insulin signaling utilizes the FOXO transcription factor to transcriptionally regulate the eukaryotic initiation factor 4e binding protein (4eBP, also known as Thor in *Drosophila*), a negative regulator of cap-dependent translation (*Gingras et al., 1999*). Importantly, our data suggest that the control of neurotransmitter release by insulin signaling is dependent on the diet conditions and likely does not reflect a role for insulin in basal neurotransmitter release. Our data supports the model that repression of FOXO activity due to insulin signaling results in reduced levels of *4eBP* mRNA, subsequent increased protein translation, and reduced SV release. The activity of 4eBP is also regulated by phosphorylation via the actions of the target of rapamycin complex (TOR) (*Beretta et al., 1996*) and numerous studies have implicated the TOR complex in the regulation of synapse function (*Costa-Mattioli et al., 2009*; *Hoeffer and Klann, 2010*; *Penney et al., 2012*; *Takei et al., 2004*; *Weston et al., 2012*). The regulation of neurotransmission by diet at the CM9 NMJ appears to be largely independent of TOR since the effect of diet on neurotransmission is not affected by rapamycin, a potent inhibitor of the TOR. Importantly, we observe that our rapamycin treatment condition does result in the predicted change in the phosphorylation state of 4eBP demonstrating that TOR can regulate 4eBP in adult Drosophila motor neurons. Because most of the data on the effects of TOR on synapse function suggest a post-synaptic role for this complex, these data suggest that the regulation of 4eBP within the CM9 motor neuron is compartmentalized with the presynaptic pool regulated specifically by FOXO and the post-synaptic pool regulated by TOR. The localization of the TOR complex in neurons is unknown, but it presumably is localized within the cytoplasm and lysosomes (*Betz and Hall, 2013*). Whether TOR is excluded from the presynaptic terminal or enriched within the postsynaptic compartment remains to be investigated.

There exist three members of the 4eBP family in mammals with 4eBP2 being the most highly expressed family member in the brain (*Banko et al., 2005*). Analysis of 4eBP2 knock-out mice has revealed that this protein is required for a broad range of cognitive and motor behaviors (*Banko et al., 2007*; *Gkogkas et al., 2013*). The changes in behavior observed in 4eBP2 knock-out mice correlate with changes in synapse function that are highlighted by changes in post-synaptic glutamate receptor function (*Banko et al., 2005*; *Bidinosti et al., 2010*; *Gkogkas et al., 2013*; *Ran et al., 2013*). To date, there is no evidence from these studies of an effect of the 4eBP2 knock-out on presynaptic function. In addition to effects on glutamate receptor function, 4eBP2 has also been implicated in the regulation of neuroligin levels (*Gkogkas et al., 2013*; *Khoutorsky et al., 2015*), a post-synaptic scaffolding protein that functions to regulate synaptogenesis and neurotransmission (*Craig and Kang, 2007*). The regulation of neurotransmission by neuroligin is likely due to its trans-synaptic interaction with the presynaptic binding protein neurexin, a cell adhesion molecule known to regulate synaptic vesicle exocytosis (*Südhof, 2008*). Thus, changes in post-synaptic neuroligin levels can result in increased presynaptic function, although this would represent a non-autonomous role for 4eBP on neurotransmitter release. It is unclear if altered neuroligin-neurexin signaling contributes to the neurotransmission phenotypes observed in the 4eBP2 knock-out mice.

We find that phosphorylation of 4eBP by dTOR has no effect on the regulation of neurotransmitter release by insulin signaling in the CM9 MN. This result suggests that there might exist separate pools of 4eBP within the neuron that specify the effects of TOR versus FOXO on synapse function. Currently, it is unclear how the compartmentalization of 4eBP activity is achieved within the pre- versus postsynaptic compartments. We have found that Staufen binds to *4eBP* mRNA in motor neurons and is required for the effects of diet on neurotransmission. In addition to mRNA transport, Staufen is also known to bind nascent mRNAs and mediate their nuclear export (*Elvira et al., 2006*; *Jansen and Niessing, 2012*; *Liu et al., 2006*; *Macchi et al., 2004*; *Miki and Yoneda, 2004*; *Miki et al., 2005*). Perhaps, the association of Staufen with nascent *4eBP* mRNAs driven by FOXO differentiates the dendritic from axonal populations of 4eBP.

In addition to *4eBP* mRNA, we also find that *Drosophila* Staufen binds strongly to *complexin* mRNA. This suggested that diet might control neurotransmission via the regulation of Complexin. In support of this model, we find that Complexin levels at the CM9 NMJ is increased in animals raised on a 2X diet compared to a 1X diet and that these levels are sensitive to changes in insulin signaling. This suggests that the increased levels of Complexin in animals raised on the 2X diet inhibit the SV release. We also find that genetically altering *complexin* levels can influence neurotransmitter release from the CM9 NMJ in a diet-dependent manner similar to what is observed with *4eBP, chico* and *dFOXO* mutants supporting the model that Complexin is an important target for the regulation of neuronal function by insulin signaling. Because both *complexin* and *4eBP* mRNAs are bound to Staufen, perhaps diet controls neurotransmitter release by altering the relative amounts of bound *4eBP* to *complexin* mRNAs.

The effects of our diet switch on neurotransmission support that an acute increase in Complexin levels can inhibit neurotransmitter release at the CM9 NMJ. It is clear from knock-out studies in mice, worms and *Drosophila* larvae that Complexin is required for normal calcium-dependent SV exocytosis and supports a facilitatory, not inhibitory, role for Complexin during neurotransmission (*Cho et al., 2010*; *Jorquera et al., 2012*; *Radoff et al., 2014*; *Reim et al., 2001*). But other studies, including acute injections and vesicle targeting studies, have indicated that Complexin can also have an inhibitory role on evoked release (*Archer et al., 2002*; *Giraudo et al., 2006*; *Liu et al., 2007*; *Ono et al., 1998*; *Tang et al., 2006*; *Tokumaru et al., 2001*). Further, comparison of *complexin* knock-down to knock-out in different neuronal cell types suggest that the effects of Complexin on SV exocytosis can be sensitive to chronic versus acute manipulations and dependent upon neuronal cell type (*Yang et al., 2013*). Although we find that diet has no effect on SV release from *Drosophila* larval NMJs, further studies will be needed to determine if this is due to differences between adult and larval motor neurons or to differences in the manipulations of Complexin. In addition, it is likely that the effects of Complexin that we observe require the co-translation of other exocytotic components. Regardless our data support the model that increases in synaptic Complexin levels resulting from insulin signaling can reduce neurotransmitter release. These results have broad implications for the effects of insulin signaling on the nervous system.

## Materials and methods

### Fly stocks

All analyses were performed on virgin female flies that were flipped to freshly made food vials every other day and kept at 50% humidity on a 12 hr light/dark cycle (*Rawson et al., 2012*). All foods were made fresh every week and flies flipped every 2 days to minimize water loss for all diet conditions. For CM9 motor neuron expression, we used the *E49-Gal4* line that was obtained from the Kristin Scott lab (*Gordon and Scott, 2009*). The *UAS-4eBP* line was obtained from the Rolf Bodmer lab (*Birse et al., 2010*). Fly lines harboring the *UAS-chico^RNAi* and the *UAS-4eBP^RNAi* transgenes were obtained from the Vienna Drosophila RNAi Center (Vienna Drosophila Resource Center, RRID:SCR_013805, stocks 101329 and 35439, respectively). The *dFOXO^94* line was obtained from Bloomington stock center (RRID:BDSC_42220) and the *dFOXO^21* line was obtained from the Marc Tatar lab (*Min et al., 2008*). The *UAS-InR^DN* fly line was obtained from the Adrian Rothenfluh lab (*Peru Y Colón de Portugal et al., 2012*). Staufen-GFP knock-in flies (GFP 311) were obtained from the Lipshitz lab (*Laver et al., 2013*). The *UAS-staufen^RNAi* line was obtained from the Bloomington stock center (RRID:BDSC_31247). All transgenes used in this study were backcrossed at lease five

generations to the $w^{1118}$ strain and rebalanced in our $w^{1118}$ background. The following genotypes were abbreviated in the text: wt = $w^{1118}$. $4eBP^{RNAi}$ = E49-Gal4/UAS-$4eBP^{RNAi}$. $chico^{RNAi}$= E49-Gal4/UAS-$chico^{RNAi}$. $staufen^{RNAi}$ = E49-Gal4/UAS-$staufen^{RNAi}$; dFOXO = $w^{1118}$; $dFOXO^{94/21}$. dFOXO, 4eBP OE = E49-Gal4/UAS-4eBP; $dFOXO^{94/21}$. 4eBP OE = E49-Gal4/UAS-4eBP.

## Dietary conditions

The low-protein (1X) and high-protein (2X) diets are exactly the same except for the amount of active yeast added and consisted of the following composition per 500 mls of food as per *Bass et al. (2007)* : 5 g agar (Genesee), 50 g active yeast (1X = 5%), or 100 g active yeast (2X = 10%) (Red Star), 25 g corn meal (Quaker), 25 g sucrose (Speckles), 1.5 ml propionic acid (Sigma), and 1.5 g tegosept (Sigma). For all experiments, newly hatched flies were kept on standard lab food for 5 days prior to being split to indicated diet conditions.

## CM9 NMJ electrophysiology

All recordings were performed in 21-day-old virgin females except of dFOXO mutants, which were feeble and died within 21 days of eclosion and therefore assayed at 14 days. Dissections and recordings were performed in a modified HL3 solution (containing, in mM: 70 NaCl, 5 KCl, 10 NaHCO3, 5 trehalose, 115 sucrose, 5 HEPES, 0.5 CaCl2, 3 MgCl2). Flies were suctioned into a Pasteur pipette and placed on top of ice for 15–20 s until the fly lost postural control. The fly was then quickly transferred to a small Sylgard dissection surface where it was decapitated. The head was moved onto its flat posterior surface, and the proboscis was then pinned into the extended position, and the entire head was covered in ice-cold dissection solution. The anterior head cuticle containing the antennae was dissected from the preparation. The proboscis was then re-pinned in the retracted position to put tension on the CM9 muscles. A loop of the lateral pharyngeal nerve was drawn into a suction electrode filled with modified HL3 (pulled glass capillary tube with a fire-polished tip, ~15 µm opening) and stimulated at 0.5–5 V for 300 µs (Digitimer Ltd., Model DS2A). The presence of a presynaptic action potential-based EPSP was verified by the presence of a distinct voltage threshold for EPSP appearance. Intracellular recordings were made on the most cranial CM9 muscle fiber accessible from the anterior side with a sharp recording electrode (~30 MΩ, filled with 3 m potassium chloride). The overall organization of the fibers is highly stereotyped from animal to animal and across age, so it is likely we are interrogating the same fiber in each recording, which is supported by our low variance. A Neuroprobe Amplifier Model 1600 (A-M Systems) was used in combination with a PowerLab 4/30 (ADInstruments, Colorado Springs, CO) to amplify and digitize the data. LabChart7 (ADInstruments, Colorado Springs, CO) was used to record the data and MiniAnalysis (Synaptosoft, Fort Lee, NJ) was used to measure both miniature EPSP (mEPSP) and EPSP events. Muscle membrane resistance was calculated using the change in muscle potential in response to current injection. Instantaneous resting membrane potential was determined by measuring the initial potential reading when the recording electrode first penetrated the muscle membrane. For hypertonic stimulation of readily releasable vesicle pools, normal recording saline was initially applied to the preparation to record baseline spontaneous activity before being replaced with recording saline supplemented with sucrose to a total final concentration of 315 mm, and recordings continued for 60 s in hypertonic saline. For diet shift experiments, animals were fed on a 1X diet from day 5-post eclosion until day 20 when half of the animals were switched to a 2X diet. For Rapamycin (Sigma, St. Louis, MO) experiments, animals were either fed a 1X diet containing either 200 µM rapamycin or vehicle control (200 µM ethanol) from 14 days post-eclosion until animals were switched onto a 2X plus rapamycin food or vehicle control for 24 hr at 20 days post-eclosion. For cycloheximide (Sigma, St. Louis, MO) experiments, animals were either fed a 1X diet containing either 35 µM cycloheximide or vehicle control (35 µM ethanol) from 19 days post-eclosion to condition the animals until they were switched onto a 2X plus cycloheximide food or vehicle control for 24 hr at 20 days post-eclosion. For all electrophysiology analyses, 7–9 animals were assayed with only one recording performed per animal (see *Table 1*). For larval analyses, eight animals were assayed with only one recording performed per animal.

## Proboscis extension reflex (PER)

Virgin flies of the appropriate genotype and dietary conditions were starved and deprived of water for 4–6 hr prior to PER analysis. Flies were anesthetized under carbon dioxide, loaded into pipet tips and allowed to recover for 30 min. For bristle tracking, digital videos of individual PERs from animals subjected to tarsal stimulation with 0.5 M sucrose were captured at 10–15 frames per second using a Zeiss MRc digital camera and Slidebook software (Intelligent Imaging Innovations, Denver CO). The Slidebook particle tracking feature was used to manually track bristles on the tip of the proboscis during PER and values for maximum velocity and average velocity were determined for each bristle path. Mean values for PER values consisted of seven animals were assayed with two PER events per animals included in analysis (14 events total).

## Antibodies and western blots

Actin reference antibody was mouse monoclonal sc-8432 used at a 1:250 dilution (Santa Cruz Biotechnology, Dallas, TX). Rabbit polyclonal antibodies against Phopho-S6K (Cell Signaling Technology Cat# 9209S RRID:AB_2269804, Danvers, MA) and Phospho-4eBP (Cell Signaling Technology Cat# 2855S RRID:AB_560835) were used at a 1:1000 dilution. For immunoblot analysis of insulin signaling, the rabbit antibody against phospho-Insulin Receptor was used at 1:1000 (Cell Signaling Technology Cat# 3021S RRID:AB_331578), the rabbit antibody against *Drosophila* Akt was used at 1:1000 (Cell Signaling Technology Cat# 9272 RRID:AB_329827), and the rabbit antibody against *Drosophila* phospho-AKT (Ser505) was used at 1:1000 (Cell Signaling Technology Cat# 4054S RRID:AB_331414). For dFOXO immunoblots, rabbit anti-dFOXO (gift of Oscar Puig Lab) was used at a 1:500 dilution. Proteins were extracted from whole flies by homogenizing them in 2x SDS sample buffer with Complete Mini protease inhibitor tablets (Roche, Indianapolis, IN) and Halt phosphatase inhibitor cocktail (Thermo Scientific, Rockford, IL). About 30 µg of denatured protein was separated on 4–15% Mini-Protean TGX Gels (Bio-Rad, Hercules, CA) until the desired band range was resolved sufficiently and transferred to nitrocellulose membranes using 350 mA with sodium tetraborate/boric acid buffer. Blocking was performed with 3% BSA for 30 min. Following incubation with primary and HRP-conjugated secondary antibodies, the blots were visualized with Novex ECL (Invitrogen, Grand Island, NY). Band intensity was quantified with ImageJ (NIH, Bethesda, MD).

## FACS and real-time PCR

For motor neuron-specific analysis, thoracic ganglion from flies expressing GFP in all motor neurons (*D42-Gal4, UAS-10X-GFP*) were removed and dissociated using 1 mg/ml collagenase (Sigma, St. Louis, MO) and sorted from non-fluorescent cells on a Beckman Dickinson Aria FACS unit using a 70 µm tip. Total RNA was extracted by FACS sorting ∼50,000 *Drosophila* neurons into RTL buffer from an RNAEasy kit, and subsequently extracted via the same kit using the standard protocol (Qiagen, Hilden, Germany). First strand cDNA was generated by reverse transcription with SuperScript III enzyme (Invitrogen, Waltham, MA). Quantitative PCR was performed on an ABI 7500 Fast Real-Time PCR system, using exon-spanning primers and SYBR green PCR premix (Applied Biosystems, Warrington, UK). The following primer pairs (Forward/Reverse, 5'-3') were used for these analyses: *4eBP*: CACTCCTGGAGGCACCA/ GAGTTCCCCTCAGCAAGCAA, *complexin*: CGCGAGAAGATGAGG-CAAGA/ CATCAGGGGATTGGGCTCTT, *tubulin*: ACAACTTCGTGTACGGACAGT/ CACCACCGAG TAGGTGTTCA, *syntaxin*: CCACAAACGGACGAGAAGACC/ CGCCGACGACTTATTCTGCT, *cacophony (cac)*: TTCGGGCGCACTGCATAAG/ GGTGGCCTTTTTCCAGGATGT. Technical replicates were performed in triplicate for all target and control genes. Transcript quantification was performed by the ΔΔCt method. For *4EBP* quantification under diet conditions, the experiment was repeated on biologically independent samples four times.

## FOXO chromatin immunoprecipitation

Chromatin immunoprecipitation (ChIP) was carried out according to Tran et al. (*Tran et al., 2012*). The thoracic ganglions from wild type flies raised on either 1X or 2X diets were dissected (n = 20 per experimental replicate), fixed, and sonicated (average fragment size ∼500 bp). Magnetic protein G beads were incubated with an anti-dFOXO polyclonal antibody (*Puig et al., 2003*) and then incubated overnight with the sonicated cell lysate. Beads incubated with rabbit pre-immune serum was used as a ChIP control. Quantitative RT-PCR was performed via the ΔΔ Ct method with primers

targeting established dFOXO response elements (FREs) in the promoter region of d4eBP (Forward 4EBP Promoter Primer: 5'- CAC CTC TTG ACT CCC AGA CAG -3'; Reverse 4EBP Promoter Primer: 5'- ATG ATA AGG GGT GTA GCG ATG -3'). Primers to a gene desert in chromosome 3 were used as reference values for normalization (Active Motif, Carlsbad CA, #71028 Drosophila Negative Control Primer Set 1).

## Staufen RNA immunoprecipitation

Staufen-GFP knock-in flies were crossed to D42, UAS-mCherry flies. The thoracic ganglia from 10 Staufen-GFP/D42, UAS-mCherry flies and control D42, UAS-mCherry flies were dissected and pooled separately. The dissected tissue was triturated at 4°C in standard homogenization buffer (250 mM sucrose, 10 mM HEPES, 1 mM EGTA, 1 mM EDTA, 0.1% NP40) with added Complete Mini Protease Inhibitor Cocktail Tablets (Roche, Penzberg, Germany) and RNAseOut RNAse inhibitors (Themo Scientific, Waltham) at 1% (v/v) concentration. This was then further ruptured by vigorous pipetting and briefly centrifuged in a standard tabletop centrifuge to pellet cell debris. The supernatant was then mixed with magnetic beads (Immunoprecipitation Kit Dynabeads Protein G, LifeTechnologies, Waltham, MA) that had previously been incubated with anti-GFP antibody (UC Davis/NIH NeuroMab Facility Cat# N86/8 RRID:AB_2313651) according to kit instructions. This was then incubated with rotation at 4°C for 4 hr. The beads were then pelleted and washed according to kit instructions. Buffer RTL Plus (with 1:100 2-mercaptoethanol) from an RNAEasy Micro Kit (Qiagen, Hilden, Germany) was added to the magnetic beads, then vortexed vigorously for 30 s. Beads were then separated via a magnetic stand, and the supernatant was used as the input to the RNAeasy kit. RNA was then isolated according to the manufacturer's instructions. RNA was stored at −80°C until used for making cDNA for qRT-PCR. Levels of RNAs bound to Staufen were compared across Staufen-GFP and control flies, normalizing to tubulin transcript.

## Immunofluorescent analysis of synaptic complexin

For these analyses, the diet and genotypes of all samples are blinded prior to the procedure. Adult *Drosophila* fly heads were pinned and dissected as previously described (*Rawson et al., 2012*) except for differences noted below. All CM9 muscles were dissected, fixed, and stained on the same day. After primary dissection, all proboscises were fixed in 4% paraformaldehyde for 15 min. After three 5-min washes in PBT, the CM9 was dissected and placed into a microcentrifuge tube containing PBT. After fixation, CM9s were blocked in ImageIt FX Signal Enhance (LifeTechnologies, Waltham, MA) for 30 min. CM9s were washed three times with PBT and subsequently placed into a new microfuge tubes for staining and incubated overnight with indicated primary antibodies. For Lateral abdominal muscles (LAMs), adult abdomens were pinned dorsal side up, filleted open, internal organs removed, and pinned down in cold dissecting saline. LAM preps were fixed in 4% paraformaldehyde for 15 min after which the LAM dissections transferred to a microcentrifuge tube and washed three times for 5 min in PBT. LAMs and CM9 preps were processed for immunofluorescence identically from this point. Both anti-cpx (rabbit polyclonal IgG, gift from Dr. Troy Littleton) and anti-Dlg (DSHB Cat# DLG1 RRID:AB_2314322) were used at a dilution of 1:500. After primary antibody incubation, preps were washed three times with PBT and placed into separate Eppendorf tubes containing a 1:500 dilution of secondary antibodies and incubated at room temperature for 1 hr. The preps were then washed three times with PBT and mounted in Vectashield containing DAPI (Vector Labs, Burlingame, CA). Dlg staining was visually inspected and any preps showing inconsistency or poor quality of Dlg staining were removed from analysis. Mounted CM9 preparations were captured using back-cooled Orca digital camera (Hamamatsu) attached to a Zeiss Axiovert immunofluorescent microscope using Slidebook software (Intelligent Imaging Innovations, Denver, CO). For intensity analysis, images were subjected to nearest neighbor deconvolution and Complexin signal intensity data acquired from sub masks generated using automated segmentation of Complexin signals provided by the Slidebook software. For the immunofluorescent analysis of Complexin, 7–11 synapses from 4 to 8 animals were analyzed. Each CM9 image yielded between 30 and 150 Complexin data points. Background masks were also generated for each synapse and values for average max pixel intensity for background subtracted from Complexin values.

## Statistical analysis

A Student's t-test was used for all pair-wise comparisons. A one-way ANOVA using a Tukey multiple comparisons test (alpha = 0.05) was used to compare all multiple values. Significance for distributions in *Figures 1* and *Figure 6* were determined using non-parametric pair-wise comparison using a Kolmogorov-Smirmov test. For all statistical analyses a confidence interval of 95% was assumed. Statistical analysis was performed using Prism6 software (Graphpad Prism, RRID:SCR_002798). The results of the statistical analyses of source data are presented in source data file.

## Acknowledgements

This work was funded by NIH/NINDS grant NS062811 to BAE and NIH/NIA training grant T32-AG021890 to RM. The Flow Cytometry Shared Resource Facility at UTHSCSA is supported by NIH-NCI P30 CA054174-20 (CTRC at UTHSCSA) and UL1 TR001120 (CTSA grant). Stocks obtained from the Bloomington Drosophila Stock Center (NIH P40OD018537) were used in this study.

## Additional information

### Funding

| Funder | Grant reference number | Author |
|---|---|---|
| National Institute on Aging | | Rebekah Elizabeth Mahoney |
| National Institute on Aging | T32-AG021890 | Rebekah Elizabeth Mahoney |
| National Institutes of Health | NS062811 | Benjamin A Eaton |
| Lawrence Ellison Foundation | AG-NS-0415-07 | Benjamin A Eaton |
| National Institutes of Health | P40OD018537 | Benjamin A Eaton |

The funders had no role in study design, data collection and interpretation, or the decision to submit the work for publication.

### Author contributions

REM, Conception and design, Acquisition of data, Analysis and interpretation of data; JA, Acquisition of data, Analysis and interpretation of data; BAE, Conception and design, Analysis and interpretation of data, Drafting or revising the article

### Author ORCIDs

Benjamin A Eaton, http://orcid.org/0000-0002-9807-5566

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
