## [Decision Letter]

[Editors’ note: this article was originally rejected after discussions between the reviewers, but the authors were invited to resubmit after an appeal against the decision.]

Thank you for submitting your work entitled "Insulin signaling controls neurotransmission via the 4eBP-dependent modification of the exocytotic machinery" for consideration by *eLife*. Your article has been evaluated by a Senior editor and three reviewers, one of whom is a member of our Board of Reviewing Editors.

Our decision has been reached after consultation between the reviewers. Based on these discussions and the individual reviews below, we regret to inform you that your work will not be considered further for publication in *eLife*.

The reviewers agreed that this is a technically demanding and potentially interesting study. However, there were questions about whether the signaling pathway is required for diet-dependent or diet-independent changes in activity that seemed difficult to resolve on the timescale of a revision. In addition, it was unclear whether the pathway is specific to proboscis MNs or a general pathway. Overall, although there was enthusiasm for the study, limitations in the current dataset hindered evaluation.

Reviewer #1:

This manuscript examines regulation of activity of proboscis motor neurons by nutritional state. The authors previously showed that diet causes changes in EPSP amplitude. In this study, they examine molecular mechanisms that regulate activity. They found that knockdown of 4eBP reduced EPSP and IRS knockdown increased EPSP. They found that complexin is increased in the high yeast diet and that manipulating complexin levels alters activity in the CM9 neurons. The authors propose that increased insulin signaling in the high yeast diet increases complexin levels and decreases neurotransmission.

There is a significant limitation in this study that makes the data difficult to interpret. Although the authors nicely show that EPSP size decreases in the 2x diet, and controls are shown for the 1x, 1x to 2x, and 2x diets, all experimental animals are shown for either the 1x or 2x diet. This makes it impossible to determine whether molecular manipulations cause a nutritional state dependent change in the response or alter responses in a state-independent manner. This is problematic. To resolve this would require examining all experimentals in the 1x, 2x and ideally 1x to 2x condition. These experiments seem unlikely to be possible to perform on the timescale of a resubmission.

Reviewer #2:

In general, I found the data presented in this paper clear and convincing. However, I have a few significant reservations about the impact and conclusions that mitigate my enthusiasm.

Impact: The authors make a case that insulin signaling plays a pathological role in neuron function in a variety of disease states, and that their model presents a possible mechanism for these effects. However, the generalizability of these findings is unclear. The fact that the same mechanism is not relevant in larval body wall motor neurons can, as the authors suggest, possibly be attributed to differences between larvae and adults. But does diet affect other adult motor neurons? Addressing this experimentally is difficult, and admittedly beyond the scope of this article. However, the fact that this particular motor neuron is involved in a very diet-sensitive behavior (feeding) makes me wonder whether this may be a specific mechanism operating in feeding motor neurons to sensitize them to decrements in nutritive intake. This would be interesting, but changes the interpretation of the data.

Conclusions: although the authors do a good job of logically dissecting the role of sequential components of the insulin pathway downstream of dietary changes, the connection between these pathway components and diet remains somewhat tenuous. To solidify this connection, I think two main points could be addressed:

1) Does the 2X diet result in measurable changes in insulin release? Related to this, the statement that the 2X diet shows that protein content is sufficient to induce the effect is inaccurate. Doubling yeast concentration in food results in increased levels of protein, carbohydrates, fats, and other nutrients. However, compensatory feeding mechanisms will likely change consumption, so it cannot be assumed that this results in increased intake of any given nutrient.

2) Currently, each genetic manipulation is only assessed in the dietary condition where it is expected to give an effect. For example, Figure 2 shows that 4eBP knockdown makes the EPSP size on the 1X diet equivalent to controls on the 2X diet. This is used to argue that 4eBP mediates the EPSP increase on the 1X diet. However, all this demonstrates is that 4eBP knockdown decreases EPSPs. The authors should show 4eBP knockdown in both dietary conditions to demonstrate that there is no diet effect on those flies.

Reviewer #2 (Additional data files and statistical comments):

The authors generally use ANOVAs to compare each genotype with the control "1X" condition. However, in cases where the argument being made is a change in the "2X" response, statistical significance should be assessed against the control 2X condition. For example, in Figure 2, *chico^RNAi^* (2X) should be shown to be significantly different from control (2X). Showing that control (2X) is different from control (1X), but *chico^RNAi^* (2X) is not, is not a robust test of the effects of *chico^RNAi^*.

Reviewer #3:

This is a very interesting study with original observations on the link between nutrient levels and synaptic function, which are currently under exploited but important in neurosciences. The main finding is that higher dietary proteins enhance insulin signaling which in turn regulates synaptic Complexin levels to reduce quantal release. Taking advantage of the available tools (mutants, RNAi, and *Gal4/UAS*) in flies the authors reached the main conclusion through careful analysis and high quality data. There were some surprises as to the roles of specific downstream players along with the insulin pathway in regulating transmitter release. However, the lack of effect by mTor is not entirely surprising as one previous study has shown that Foxo rather than mTor plays a critical role in neuronal excitability. Overall, this study should be of interest to all neuroscientists interested in synaptic plasticity, cell biologists in cell signaling, and clinicians in diabetes and aging.

I do have some questions and concerns:

1) I am puzzled by the lack of conservation of this signaling pathway at the larval NMJ. One can imagine the critical role of insulin signaling in coordinating rapid larval growth and synaptic function. Yet this pathway as described by the authors is not used in larvae. However, this issue needs some reconsideration. The author ignored the landmark study by Mike Stern's lab (Howlett et al., PLoS Genet 4(11): e1000277), in which they showed that Foxo and mTor indeed had similar effects on larval NMJ function as the current authors found with adult fly NMJs. The Stern paper should be cited! Additionally, it might be a good idea to use the Stern approach to re-examine the role of nutrients and insulin signaling at larval NMJs.

2) Neuronal excitability vs. SNARE-mediated fusion: the authors have good data indicating the involvement of Complexin levels at the adult NMJ as a downstream factor of nutrient/insulin signaling. Without much doubts Complexin is known to be a regulator of SNARE-mediated fusion. One also wonders, given the Stern study, whether nutrients also regulate neuronal excitability.

In Figure 6, the authors used confocal imaging to quantify Complexin levels. Is this method quantitative enough or without bias? Is the postsynaptic marker Dlg used as a control to normalize Complexin signal levels?

3) Because the authors did not observe an effect of nutrients at the larval NMJ, I wonder how universal it is for insulin signaling to regulate transmitter release at other adult synapses (e.g., central synapses as well as another NMJ)?

I am not interested in asking the authors to do more additional work, but it is important for them to demonstrate that this pathway is physiologically conserved. Thus, data from either larval NMJ or another adult synapse would strengthen the significance of their new findings.

Reviewer #3 (Additional data files and statistical comments):

Supplementary figures are appropriate.

---

## [Author Response]

[Editors’ note: the author responses to the first round of peer review follow.]

Reviewer #1:

There is a significant limitation in this study that makes the data difficult to interpret. Although the authors nicely show that EPSP size decreases in the 2x diet, and controls are shown for the 1x, 1x to 2x, and 2x diets, all experimental animals are shown for either the 1x or 2x diet. This makes it impossible to determine whether molecular manipulations cause a nutritional state dependent change in the response or alter responses in a state-independent manner. This is problematic. To resolve this would require examining all experimentals in the 1x, 2x and ideally 1x to 2x condition. These experiments seem unlikely to be possible to perform on the timescale of a resubmission.

We agree with the reviewers that the lack of certain diet conditions for specific mutant backgrounds in the previous data set limited our ability to exclude a general role for these insulin signaling pathway components during neurotransmitter release. This is particularly relevant for 4eBP and, which are believed to have highly pleiotropic functions within the cell. The reviewers suggest investigating the effects of *4eBP* mutations and others on multiple diet conditions to resolve whether this signaling pathway is required specifically for the dietary response, or if this signal plays a more general role during SV release. To address these questions, we have included new electrophysiological data for proximal (4eBP, *chico*, and) and distal components (Complexin) of the signaling pathway on multiple diets. To support this analyses, we have also performed immunoblot analysis to support that insulin signaling is increased under our 2X and 1-2X diet shift conditions compared to our 1X diet condition (Figure 2). These data also suggest that the 2X condition and the 1-2X shift both result in qualitatively similar effects on insulin signaling. This new data is now included in Figure 2 and in Table 1. We will discuss these new data in more detail below.

We do not believe that these analyses are required for all experimental conditions. The 1-2X diet shift allows us to acutely increase the insulin signaling allowing us to design ideal paradigms for investigating the role of TOR and Staufen during the propagation of the insulin signal. Given our new data, we don’t believe that including new diet conditions for these experiments is necessary, especially given the similarities between the 2X and 1-2X diet conditions in terms of functional effects and whole animal insulin signaling.

1) 4eBP: Previously we showed that knock-down of 4eBP in animals raised on a 1X diet reduced release to levels observed in 2X controls. We have now repeated this analysis and have included data for knock-down of *4eBP* in both 2X and 1X-2X shift animals. The amount of neurotransmitter released in *4eBP* knock-down flies raised on either 2X and 1X-2X shift is the same as the 1X *4eBP* knock-downs and the 2X control animals (Figure 2). This new data provides support for the model that 4eBP is specifically required for the effects of diet on neurotransmitter release. It also suggests that 4eBP is not required for basal neurotransmission, similar to what we observe at the larval NMJ. Taken together, these data support that under high insulin signaling, 4eBP activity is low and contributes very little to neurotransmitter release. In addition to a brief presentation of the data in the results we have included a brief discussion of these data in the Discussion.

2) Chico and d: We also performed a similar analysis of *chico* mutants and observe that knock-down of *chico* in animals raised on a 1X diet resulted in a significant increase in release compared to 1X controls. This was also observed for *chico* mutants subjected to a 1-2X diet shift. These results suggest that there is basal insulin signaling through Chico even under the 1X diet conditions, which is supported by faint insulin receptor phosphorylation under 1X conditions. This is also consistent with our observation that overexpression of 4eBP in wild type animals raised on 1X diet increases neurotransmitter release compared to 1X control animals (Figure 4). We have included a statement in the results describing these data. We have also analyzed *d* mutants and found that consistent with the above data, *d* mutants on 2X diets have neurotransmitter release similar to 2X controls.

3) Complexin: We have now included data demonstrating that SV release in animals expressing Complexin is the same in animals raised on 2X compared to animals raised on 1X. Taken together, these new data support our model that this signaling system functions to control neurotransmitter release in response to changes in diet.

Reviewer #2:

In general, I found the data presented in this paper clear and convincing. However, I have a few significant reservations about the impact and conclusions that mitigate my enthusiasm.

Impact: The authors make a case that insulin signaling plays a pathological role in neuron function in a variety of disease states, and that their model presents a possible mechanism for these effects. However, the generalizability of these findings is unclear. The fact that the same mechanism is not relevant in larval body wall motor neurons can, as the authors suggest, possibly be attributed to differences between larvae and adults. But does diet affect other adult motor neurons? Addressing this experimentally is difficult, and admittedly beyond the scope of this article. However, the fact that this particular motor neuron is involved in a very diet-sensitive behavior (feeding) makes me wonder whether this may be a specific mechanism operating in feeding motor neurons to sensitize them to decrements in nutritive intake. This would be interesting, but changes the interpretation of the data.

We would like to first point out that our analysis of *4eBP* mRNA expression in response to diet was generated on motor neurons isolated from thoracic ganglion demonstrating that the regulation of 4eBP by diet is common to all motor neurons. This point has been strengthened in the text. We also now provide evidence of changes in Complexin levels at the lateral abdominal muscle NMJs in response to diet (Figure 6—figure supplement 1). These new data as well as our previous RT-PCR data support that this signaling system is not only a CM9 MN phenomenon but is likely generalized to adult motor neurons.

Finally, I think that it is important to remember that changes in peripheral insulin has important effects on brain function but that the cellular mechanisms, especially the regulation of presynaptic function, have not been pursued in mammals. Thus, the conservation of this effect of insulin is unknown. But the signaling pathway components and the SV release mechanisms detailed in this manuscript are highly conserved with mammals so I would argue that it is possible that some aspect of this regulation will be conserved.

*Conclusions: although the authors do a good job of logically dissecting the role of sequential components of the insulin pathway downstream of dietary changes, the connection between these pathway components and diet remains somewhat tenuous. To solidify this connection, I think two main points could be addressed:*

1) Does the 2X diet result in measurable changes in insulin release? Related to this, the statement that the 2X diet shows that protein content is sufficient to induce the effect is inaccurate. Doubling yeast concentration in food results in increased levels of protein, carbohydrates, fats, and other nutrients. However, compensatory feeding mechanisms will likely change consumption, so it cannot be assumed that this results in increased intake of any given nutrient.

There exist a number of publications demonstrating the effects of diet on *Drosophila* insulin-like peptide (Dilps) gene expression (Morris et al., 2012; Gronke et al., 2010) but there is no assay for measuring circulating Dilps levels in *Drosophila*. To provide evidence of increased Dilp signaling, we used biochemical analyses of insulin receptor phosphorylation from flies raised on our different diet conditions. The increase in the phosphorylation state of the insulin receptor reflects an increase in insulin binding to the receptors providing strong evidence of increased insulin signaling under our diet conditions. We also observe an increase in Akt phosphorylation, consistent with increased insulin signaling (Bai et al., 2015 (Tatar lab); Bjedov et al., 2010 (Partridge lab)).

2) Currently, each genetic manipulation is only assessed in the dietary condition where it is expected to give an effect. For example, Figure 2 shows that 4eBP knockdown makes the EPSP size on the 1X diet equivalent to controls on the 2X diet. This is used to argue that 4eBP mediates the EPSP increase on the 1X diet. However, all this demonstrates is that 4eBP knockdown decreases EPSPs. The authors should show 4eBP knockdown in both dietary conditions to demonstrate that there is no diet effect on those flies.

See response to Reviewer #1.

Reviewer #2 (Additional data files and statistical comments):

The authors generally use ANOVAs to compare each genotype with the control "1X" condition. However, in cases where the argument being made is a change in the "2X" response, statistical significance should be assessed against the control 2X condition. For example, in Figure 2, chico^RNAi^ (2X) should be shown to be significantly different from control (2X). Showing that control (2X) is different from control (1X), but chico^RNAi^ (2X) is not, is not a robust test of the effects of chico^RNAi^.

We agree and have reanalyzed this data as requested.

Reviewer #3:

This is a very interesting study with original observations on the link between nutrient levels and synaptic function, which are currently under exploited but important in neurosciences. The main finding is that higher dietary proteins enhance insulin signaling which in turn regulates synaptic Complexin levels to reduce quantal release. Taking advantage of the available tools (mutants, RNAi, and Gal4/UAS) in flies the authors reached the main conclusion through careful analysis and high quality data. There were some surprises as to the roles of specific downstream players along with the insulin pathway in regulating transmitter release. However, the lack of effect by mTor is not entirely surprising as one previous study has shown that Foxo rather than mTor plays a critical role in neuronal excitability. Overall, this study should be of interest to all neuroscientists interested in synaptic plasticity, cell biologists in cell signaling, and clinicians in diabetes and aging.

I do have some questions and concerns:

1) I am puzzled by the lack of conservation of this signaling pathway at the larval NMJ. One can imagine the critical role of insulin signaling in coordinating rapid larval growth and synaptic function. Yet this pathway as described by the authors is not used in larvae. However, this issue needs some reconsideration. The author ignored the landmark study by Mike Stern's lab (Howlett et al., PLoS Genet 4(11): e1000277), in which they showed that Foxo and mTor indeed had similar effects on larval NMJ function as the current authors found with adult fly NMJs. The Stern paper should be cited! Additionally, it might be a good idea to use the Stern approach to re-examine the role of nutrients and insulin signaling at larval NMJs.

Although *chico* signaling has been previously implicated in synaptogenesis within the adult fly brain (Martin-pena et al. 2006), a role for *chico* and *4eBP* during neurotransmission at the larval NMJ has not been reported. We have no reason to doubt our experiments and we have repeated these experiments multiple times over the course of numerous years (see Table 1). We are also aware of a competing manuscript describing a postsynaptic role for *4eBP* in the larval muscle during homeostatic synaptic plasticity. 4eBP is likely the postsynaptic target of the TOR complex in the larval muscle that has previously been implicated in retrograde signaling during homeostasis at the larval NMJ (Penney et al., 2014). These researchers also found no role for 4eBP in the motor neuron during neurotransmission or synaptic homeostasis (personal communication) consistent with our data.

Although there is no evidence for *4eBP* or *chico* at the larval NMJ, there is previous data from the larval NMJ demonstrating a requirement for *d* during the control of neurotransmission by PI3K signaling downstream of mGluR signaling (Howlett et al., PLoS Genet 4(11): e1000277). Our data demonstrate that *d* is required for neurotransmission at the CM9 NMJ in animals raised on 1X diet. Although it is clearly difficult to compare these experiments, there does appear to be a requirement for *d* function during neurotransmission at both larval and adult NMJs. Finally, we appreciate the reviewer pointing our omission of this important study of *d* function during neurotransmission and have included it in our current manuscript.

2) Neuronal excitability vs. SNARE-mediated fusion: the authors have good data indicating the involvement of Complexin levels at the adult NMJ as a downstream factor of nutrient/insulin signaling. Without much doubts Complexin is known to be a regulator of SNARE-mediated fusion. One also wonders, given the Stern study, whether nutrients also regulate neuronal excitability.

In this paper, the Stern lab utilize the onset of long-term facilitation (LTF) as a reporter of motor neuron excitability largely based on previous studies of the effects of ion channel mutants on this response. In this study, the authors show that is required for normal basal neurotransmitter and neuronal excitability (as measured by LTF induction) downstream of mGluR signalling. Unfortunately, these researchers state that they use an undefined diet so there is no information given about nutrient conditions in this paper. Regardless, these data support that is required for basal neurotransmitter release at both larval and adult NMJs. We have included this point in the new manuscript.

The reviewer suggests using the Stern approach (we assume induction of LTF) to investigate if nutrients also influence excitability to the CM9 motor neuron. We agree that this is an interesting question but beyond the focus of this paper, which are the mechanisms downstream of that control neurotransmission. It should be noted that there is very little known about the mechanisms downstream of that alter neuronal function or excitability. Our current story represents one of the first descriptions of downstream mechanisms from that alter neuronal function.

In Figure 6, the authors used confocal imaging to quantify Complexin levels. Is this method quantitative enough or without bias? Is the postsynaptic marker Dlg used as a control to normalize Complexin signal levels?

We agree that this is a tenuous approach to determining changes in synaptic protein levels but we have performed these as carefully as possible. We have attempted to use immunoblot analysis of dissected proboscis but have been unable to generate enough material to get a signal with the Cpx antibody. That said, this is a direct method for measuring synaptic protein levels that we support with our genetic manipulations of Complexin levels. We would also like to point out that all of the analyses of fluorescent levels were done in a blinded fashion and identities of genotypes only revealed after analysis. We have clarified this in the Methods. In addition, during our development of this approach we analyzed synaptic Complexin levels at the CM9 NMJ in *wt* versus *cpx/+* that revealed a significant decrease in synaptic protein levels. Importantly, these results also correlated well with the functional data from the recordings in the *cpx/+* heterozygotes. We have included these data in Figure 6 of this version of the manuscript to strengthen this approach. We do not normalize to Dlg in these analyses. Rather, Dlg is used to evaluate the quality of the staining of each preparation. Preparations in which the Dlg staining is inconsistent are discarded prior to analysis of Cpx levels. Cpx signals are not normalized but are corrected for background of the Cpx staining (which is low, see Figure 6).

3) Because the authors did not observe an effect of nutrients at the larval NMJ, I wonder how universal it is for insulin signaling to regulate transmitter release at other adult synapses (e.g., central synapses as well as another NMJ)?

I am not interested in asking the authors to do more additional work, but it is important for them to demonstrate that this pathway is physiologically conserved. Thus, data from either larval NMJ or another adult synapse would strengthen the significance of their new findings.

See first response to Reviewer #2.